



# Odd hydrogen response thresholds for indication of solar proton and electron impact in the mesosphere and stratosphere

Tuomas Häkkilä[1], Pekka T. Verronen[1,2], Luis Millán[3], Monika E. Szeląg[1], Niilo Kalakoski[1], and Antti Kero[2]

[1]Space and Earth Observation Centre, Finnish Meteorological Institute, Helsinki, Finland
[2]Sodankylä Geophysical Observatory, University of Oulu, Sodankylä, Finland
[3]Jet Propulsion Laboratory, California Institute of Technology, Pasadena, CA, USA

**Correspondence:** Tuomas Häkkilä (tuomas.hakkila@fmi.fi)

**Abstract.** Understanding the atmospheric forcing from energetic particle precipitation (EPP) is important for climate simulations on decadal time scales. However, presently there are large uncertainties in energy-flux measurements of electron precipitation. One approach to narrow these uncertainties is by analyses of EPP direct atmospheric impacts and their relation to measured EPP fluxes. Here we use odd hydrogen observations from the Microwave Limb Sounder and Whole Atmosphere Community

Climate Model simulations, together with EPP fluxes from the GOES and POES satellites, to determine the response thresholds to solar proton events (SPEs) and radiation belt electron (RBE) precipitation. We consider a range of altitudes in the middle atmosphere, and all magnetic latitudes from pole to pole. We find that the lower flux limits for day-to-day EPP impact detection using OH and $HO_2$ are of the order of $10^2$ protons $cm^{-2}s^{-1}sr^{-1}$ ($E > 10$ MeV) and $10^4$ electrons $cm^{-2}s^{-1}sr^{-1}$ ($E = 100$–300 keV). Based on the simulations, nighttime OH and $HO_2$ are good EPP indicators in the polar regions, and provide

best coverage in altitude and latitude. Due to larger background concentrations, daytime detection requires larger EPP fluxes and is possible in the mesosphere only. SPE detection is easier than RBE detection because a wider range of polar latitudes is affected. We also find that MLS OH observations indicate a clear nighttime response to SPE and RBE in the mesosphere, similar to the simulations, while $HO_2$ data are overall too noisy for confident EPP detection.

## 1 Introduction

Solar energetic particle precipitation (EPP) affects the polar atmospheric chemistry directly at the altitude region from upper stratosphere to lower thermosphere. Ionization caused by precipitating protons and electrons leads to, e.g., production of odd hydrogen and odd nitrogen from ionic reactions, and subsequently to loss of ozone through catalytic reactions (Sinnhuber et al., 2012). There is evidence of EPP-driven variability in winter-springtime ozone (Andersson et al., 2014a; Damiani et al., 2016), which could further connect to decadal variability of regional climate via modulation of polar vortex dynamics and the

top-down coupling (e.g. Seppälä et al., 2014).

In atmospheric and climate modeling, EPP forcing can be defined using satellite-based particle flux observations (Matthes et al., 2017, and references therein). Solar wind proton fluxes are continuously measured by detectors aboard the GOES satellites in the geosynchronous orbit (https://www.ngdc.noaa.gov/stp/satellite/goes/). These measurements provide a good repre-





sentation of proton forcing because MeV protons have enough rigidity, i.e. momentum/charge, to penetrate through Earth's magnetic field in polar regions and enter the atmosphere directly from the solar wind. Several studies have shown that the observed atmospheric effects can be well represented in models using the GOES proton observations if the relevant ion-neutral chemistry is considered as well (Jackman et al., 2001; Verronen et al., 2006; Funke et al., 2011; Andersson et al., 2016). For

electron precipitation the situation is different. Electrons have less mass than protons and are captured by Earth's magneto-sphere, e.g. in the radiation belts (Baker et al., 2018), from where they are eventually lost either to space or into the atmosphere. Satellite-based observations of electron precipitation fluxes are being made from low-orbiting satellites but these measurements do not capture full spatio-temporal varibility, and also suffer from restricted measurement geometry and proton contamination, like in the case of the MEPED/POES instruments (Rodger et al., 2010a; van de Kamp et al., 2018). Atmospheric impacts seen

in observations seem to indicate a need for a large adjustment of the electron forcing representation in simulations (Clilverd et al., 2012; Randall et al., 2015). Thus atmospheric observations of EPP impact could help to understand the uncertainties in the electron flux data and how these flux observations relate to effects in the atmosphere (e.g. Verronen et al., 2011). Thus EPP detection limits for atmospheric data are valuable information.

Ground-based ionospheric observations provide the most direct measure of EPP atmospheric impact (e.g. Verronen et al.,

2015; Heino et al., 2019), but in practice only satellite-based measurements can offer a global view. Measurement of odd hydrogen species (OH, $HO_2$) are well-suited for monitoring EPP impacts due to their relative short chemical lifetime (Verronen et al., 2006; Damiani et al., 2010). Satellite-based observations of OH were made continuously in 2004–2009 by the Microwave Limb Sounder (MLS) aboard the Aura satellite (Pickett et al., 2006; Pickett et al., 2006; Pickett et al., 2008). They have since been used to study both solar proton events and electron precipitation (Verronen et al., 2006; Damiani et al., 2010; Jackman

et al., 2011; Andersson et al., 2012; Verronen et al., 2013; Jackman et al., 2014; Andersson et al., 2014b), particularly at 60–80 km altitudes where largest EPP events can produce order-of-magnitude increases. Compared to the OH observations, MLS $HO_2$ measurements are being made since 2004 and thus provide a longer timeseries than the OH observations. However, the $HO_2$ data from operational processing have a lower signal to noise ratio and only cover the lower mesosphere and stratosphere (Pickett et al., 2008; Livesey et al., 2018). Thus the use of standard $HO_2$ data has been limited to large proton events (Jackman

et al., 2011; Zou et al., 2018). An improved offline processing of $HO_2$ data provides better quality (Millán et al., 2015), extending the altitude range to the mesopause and enhancing possibilities for studies of daily EPP impact.

In this paper, we use MLS observations of OH and $HO_2$ to determine EPP flux thresholds for impact detection in the stratosphere and mesosphere. Looking at different latitudes bands and altitudes individually, we first consider large proton events, which have a well-known flux from satellite observations, to develop a method for EPP threshold detection. We then

apply the same method to medium-energy electrons for which the satellite based flux observations are not all-inclusive. We compare satellite-data-based thresholds to those from the Whole Atmosphere Community Climate Model (WACCM, version 4), to discuss both limitations of the satellite data and EPP forcing in WACCM. Our results provide the limits of MLS OH and $HO_2$ for EPP detection and usability in EPP studies.



## 2 Data and models

The microwave limb sounder (MLS) measures millimeter and submillimeter thermal emission from the Earth's limb atmosphere, from which temperature, trace gases, and cloud ice are retrieved. Launched in July 2004 into a Sun-Synchronous near-polar orbit, the geographic latitude coverage of the measurements is from $82°$S to $82°$N on each orbit and measurements
are made during both day and night conditions. The instrument is described in detail by Waters et al. (2006). A detailed description of MLS version 4 data products and quality is given by Livesey et al. (2018). The MLS target species in the stratosphere and mesosphere include OH and $HO_2$.

For the version 4 OH data used in our study, the recommended pressure range of observations is 32–0.0032 hPa (approx. altitudes 25–95 km). At levels with $p > 0.01$ hPa, the vertical resolution of observations is 2.5 km. Version 2.2 data has been
validated with balloon-borne remote-sensing instruments and with ground-based column measurements (Pickett et al., 2008; Wang et al., 2008). Instead of the standard $HO_2$ data, we use data from the offline processing described in detailed and validated against other observations by Millán et al. (2015). This algorithm retrieves daily zonal means of $HO_2$ over an extended vertical range by first averaging the radiances in $10°$ bins. The recommended pressure range for the offline $HO_2$ data is 10–0.0032 hPa ($\approx$35–90 km), vertical resolution is about 4 km between 10 and 0.1 hPa, 8 km at 0.02 hPa and around 14 km for lower
pressures. Day and nighttime data are provided separately using measurement tangent point solar zenith angle limits $< 90°$ and $> 100°$, respectively.

The Whole Atmosphere Community Climate Model (WACCM) is a chemistry-climate model that extends from the surface to about $5.9 \times 10^{-6}$ hPa ($\approx$140 km) with horizontal resolution $1.9°$ latitude by $2.5°$ longitude. WACCM physics and atmospheric response to solar and geomagnetic forcing variations are described by Marsh et al. (2007). Details about WACCM version 4
coupled simulations and overview of the model climate can be found in Marsh et al. (2013). Here, we utilize WACCM-D, a variant of WACCM version 4 in which standard parametrization of EPP-driven odd hydrogen and odd nitrogen production is replaced by a set of D-region ion chemistry reactions (Verronen et al., 2016), and which reproduces better the observed OH response during EPP (Andersson et al., 2016). We used WACCM-D in the specified dynamics scenario (SD-WACCM-D) forced with meteorological fields (temperature, winds, surface pressure) from Modern-Era Retrospective analysis for Research and
Applications (MERRA, Rienecker et al., 2011). The daily atmospheric ionization rates due to 1–300 MeV solar protons are calculated based on flux data available from the National Oceanic and Atmospheric Administration (NOAA) Space Environment Center (http://sec.noaa.gov/Data/goes.html) and the methodology discussed in Jackman et al. (2009). These are applied at geomagnetic latitudes $> 60°$ in both hemispheres. The daily zonal mean ionization rates at geomagnetic latitudes $40°$–$72°$ from precipitating 30–1000 keV electrons are taken from the APEEP proxy model which is recommended for the Coupled
Model Intercomparions Project (van de Kamp et al., 2016; Matthes et al., 2017). We carried out a simulation for a period between 2000–2012 which covers the whole period of MLS OH observations. WACCM-D output, including OH and $HO_2$, was saved at MLS measurement times and locations to ensure one-to-one match between the model and the measurements in the analysis.


EPP forcing patterns follow the geomagnetic latitudes, rather than the geographic ones, and the odd hydrogen response is expected to show similar patterns due to its short chemical lifetime (e.g. Andersson et al., 2018). Thus we use a modified version of the $HO_2$ offline algorithm where the radiances were averaged using geomagnetic latitudes instead of geographic latitudes (for a definition of Altitude-Adjusted Corrected Geomagnetic Coordinates, see e.g. Shepherd, 2014). These geomagnetic latitudes

are shown in Fig. 1. Before the analysis, the MLS OH and WACCM-D data were pre-processed the same way as it was done for $HO_2$. The data were divided into daily daytime and nighttime sets using the same solar zenith angle limits and averaged zonally after sorting observations into $10°$ geomagnetic latitude bins, running from $-85°$ through to $+85°$. Through this manuscript these bins are referred to by their central latitude, i.e. the latitude $40°$ refers to the zonal average from geomagnetic latitudes $35°-45°$. It should be noted that due to separating the data both by latitude and by day/night conditions, there is sometimes

little data available at the highest latitudes. For example, sometimes there is no nighttime data in June for latitude $80°$. The daily averages of existing data points at the high geomagnetic latitude bins may also be skewed towards lower geographic latitudes. However, this should not affect our comparisons because WACCM-D was sampled at the MLS times and locations and binned in the same manner.

In the analysis, we use the pressure level grid from the MLS measurements, as sections of the MLS $HO_2$ and OH pressure

grids are the same. Hence the overlapping sections of MLS $HO_2$ and OH values were selected, whereas the least-squares interpolation method recommended by Livesey et al. (2018) was used to convert WACCM-D data onto the MLS grid. The same pressure grid allows for direct comparisons between the data sets. The MLS $HO_2$ offline data are provided in both mixing ratios (ppmv) and concentrations (molec./$cm^3$). Thus daily total density profiles can be calculated by dividing $HO_2$ concentrations by $HO_2$ mixing ratios. To ensure consistency between the data sets, MLS OH and WACCM-D data were

converted from mixing ratios to concentrations using the same total densities from MLS $HO_2$. For the general comparisons, the MLS HO2 averaging kernels were applied to the WACCM-D HO2 fields. As most of the averaging kernels are fairly similar, only the daytime and nighttime kernels corresponding to magnetic equator were used and applied to all latitude bins. MLS averaging kernels were not used on WACCM-D OH data because the effect was expected to be small. In the threshold detection, the MLS averaging kernels were not applied on WACCM-D $HO_x$ data.

In order to show the connection between EPP and odd hydrogen changes, we use proton flux measurements from the Geostationary Operational Environmental Satellite (GOES-11, https://www.ngdc.noaa.gov/stp/satellite/goes/) as a measure of proton precipitation over the polar caps. Daily average fluxes of protons with energies greater than $10$ MeV are used as an indicator because those directly affect atmospheric altitudes below about $90$ km (for the relation between EPP energies and penetration altitudes, see Turunen et al., 2009, Figure 3). As a measure of electron precipitation, we use flux observations

from the Medium Electron Proton and Electron Detector aboard the Polar Orbiting Environmental Satellite (Evans and Greer, 2004). Daily average electron fluxes from an energy range between $100$ keV and $300$ keV, from the $50°$ magnetic latitude bin, are used as indicator of radiation belt electron precipitation (RBE) affecting mesospheric altitudes. While the magnitude of RBE forcing will depend on latitude more than for the proton forcing, the $50°$ latitude bin was selected because of its most pronounced atmospheric $HO_x$ impact (e.g. Verronen et al., 2011). The assumption here is that the RBE forcing will

vary similarly with magnetic activity across affected latitudes although the magntitude can differ. The electron flux data are





exclusively from the zero-degree telescope and have been pre-processed to remove known quality issues (for a description, see Verronen et al., 2011, and references therein), e.g. data suffering from proton contamination have been excluded using the methods described by Rodger et al. (2010a).

## 3  Methods

First, we present an overall comparison between WACCM-D and MLS odd hydrogen data. We use monthly average concentrations to identify the main similarities and differences between the data sets, focusing on the shape, strength, and location of concentration peaks. Comparisons are done using time series of monthly average concentrations as well as individual months. We selected individual months from the year 2009 due to it being a low EPP activity period, thus providing relatively EPP-free comparison of the background conditions. These comparisons provide us information on the overall representation of odd hydrogen in WACCM-D, which can help to understand the differences in EPP detection thresholds.

Starting from the approach of Verronen et al. (2011), we use daily average data in EPP detection. However, Verronen et al. (2011) studied only four selected RBE events. And although they demonstrated the correlation between EPP and OH, they did not use $HO_2$ data or pursue the detection limits. Before determining threshold values for the SPE and RBE detection, daily climatologies of OH and $HO_2$ concentrations were removed from the odd hydrogen data. Climatological values were calculated for each day of the year by first calculating daily average OH and $HO_2$ night and day concentrations of available data at each grid point, separately for WACCM-D and MLS. A nine-day moving average was then calculated to smooth the time series and produce the final climatologies. After the climatology is removed, the background concentrations do not display seasonal variability, which makes it possible to combine EPP event periods from different seasons for the threshold detection. Examples of this de-seasonalising effect can be seen in Fig. 2.

For the SPE threshold determination, we selected data from all months during which the daily proton flux indicates an event, i.e. it exceeds the limit of 10 $protons\,cm^{-2}s^{-1}sr^{-1}$, see Fig. 3 (upper panel). All months used in the analysis are listed in Table 1. Using data from the selected months, the SPE flux values were plotted against the climatology-free odd hydrogen concentrations (i.e. anomalies). A first degree polynomial was then fitted to the data (linear fit). A limit for a significant SPE-driven enhancement in odd hydrogen concentration was calculated by adding half of the standard deviation of the concentrations to the median concentration. Since the concentrations are de-seasonalised, the median is expected to be close to zero. The addition of half of the standard deviation guarantees that a concentration anomaly of one standard deviation from the climatology is detected. The SPE flux threshold value is found at the point where the linear fit intersects this limit. Examples of the SPE threshold determination are shown in Fig. 4 (left panel). This process is applied for each latitude bin and at each pressure level separately. To identify those thresholds that are reasonable, they are filtered using correlations between the SPE fluxes and odd hydrogen concentrations. All thresholds with corresponding correlation coefficient $\geq 0.35$ are accepted as reasonable and the rest are discarded. This filtering effectively removes threshold values at lower latitudes where SPE impact is not expected.

The RBE threshold values are determined using the same method but with different RBE flux and correlation limits. RBE months were found using an RBE flux limit of $1.5 \times 10^4$ $electrons\,cm^{-2}s^{-1}sr^{-1}$ (see Fig. 3, lower panel), and data from





these months are used in the analysis. However, the months having an SPE event, as defined in the previous paragraph, were excluded. The RBE fluxes from MEPED are not reliable durings SPEs (Rodger et al., 2010a), and SPE forcing would likely interfere with the RBE threshold detection. Indeed, exclusion of SPE months leads to stronger correlations between the RBE flux and odd hydrogen concentrations (not shown). For a full list of the months used in the analysis, refer to Table 1. As

with SPEs, RBE thresholds were found using a linear fit, the median and half standard deviation of odd hydrogen data (see Fig. 4, right panel). For RBE events, threshold values with corresponding flux-concentration correlation coefficients $\geq 0.25$ are accepted as reasonable.

As seen in Fig. 4 (left panel) at 0.1–0.2 $\mathrm{protons\ cm^{-2}s^{-1}sr^{-1}}$, in some cases there is a large number of data points at low fluxes. To find out the impact to our analysis, we performed tests where we excluded these low-flux points. In general,

correlations between flux data and $HO_x$ concentrations get stronger, but thresholds increase due to larger standard deviation. These effects would, however, not change our conclusions.

## 4   Results

### 4.1   Overall comparison between WACCM-D and MLS

In general, WACCM-D and MLS compare reasonably well in the magnitude and spatio-temporal variability of OH and $HO_2$.

In daytime concentration profiles, there is a maximum in the stratosphere and mesosphere, which reflects the production being dependent on atomic oxygen and Lyman-alpha radiation, respectively. MLS and WACCM-D both show these maxima with largest values seen in the lower latitudes, as seen in Fig. 5 which shows a typical case (April 2009). There are general differences as well. At the equator, the mesospheric concentration peaks are typically stronger in MLS data, and for OH also the peak altitude is higher by around 2.5 km than in WACCM-D. On the other hand, the stratospheric peaks are stronger in

WACCM-D, at least based on OH concentrations. The noisiness of MLS $HO_2$ data is clear at lower atmospheric levels, as seen in Fig. 5. At the polar regions, where EPP impact is expected, absolute differences are smaller while WACCM-D shows generally larger OH concentrations. Thus, in this month, the EPP detection from MLS OH data could be easier due to the lower background concentration in the mesosphere.

The daytime similarities and differences are also visible in the time series of monthly average concentrations, shown for

magnetic latitude 70°N in Fig. 6. Maximum peak values are seen in the summer months in both WACCM-D and MLS. WACCM-D shows a stronger stratospheric OH peak by 10–20% and a weaker mesospheric OH peak by up to a factor of two. In WACCM-D, the vertical transition between the OH summertime peaks is more continuous, while in MLS data there is a clear minimum between them around 0.1 hPa. EPP detection in the summer mesosphere is likely harder from MLS than WACCM-D because of larger background concentrations. Again, the MLS $HO_2$ data become noisy at the lower atmospheric

levels Note that the $HO_2$ data could be improved below the 1 hPa level by using the day-night difference (Livesey et al., 2018), but this was not done in our analysis. The mesospheric summertime peak is stronger in MLS by some tens of percent and is located a few km lower than in WACCM-D. In wintertime, both WACCM-D and MLS show lower OH values than in summer, particularly in the mesospheric altitudes, while the altitude distributions and concentrations are very similar.





Nighttime concentrations show some similarities and differences as well, a typical example is presented in Figure 7 from latitude $60°$S. Overall, nightly concentrations are low in both MLS and WACCM-D. MLS $HO_2$ is a clear exception at lowest atmospheric levels, with much larger concentrations than in WACCM-D. Note, however, that the nighttime $HO_2$ data are only valid from 1 to $0.0032$ hPa, while from 10 to 1 hPa the concentrations are close to zero and only used to correct the daytime

observations. Thus, these high values are likely an artefact of the low signal-to-noise quality of the observations.

Both WACCM-D and MLS show the characteristic OH peak in the mesosphere, produced at night from reaction between atomic hydrogen and $O_2$. The peak is approximately at the same pressure level of $0.01$ hPa, but its magnitude is larger and the seasonal variability is weaker in WACCM-D. In summer, the peak occurs at a higher altitude and is stronger in both WACCM-D and MLS, although the altitude variation is slightly less obvious in MLS data. The MLS OH data also shows oscillating

minima and maxima in summertimes at $0.1$–$0.01$ hPa, but these may be an artefact of the observations and are not seen in WACCM-D.

For $HO_2$, WACCM-D shows an upper mesospheric peak with a clear seasonal cycle, coinciding with the OH peak. In addition, there is another clear peak at around $0.5$ hPa, with a seasonal cycle: a minimum in winter, and clear maxima in spring and autumn. MLS $HO_2$ data are noisy, and there is no clear patterns of seasonal variability.

Our WACCM-D/MLS $HO_2$ comparison results are similar to those by Millán et al. (2015), although it should be noted that they presented the data in geographic coordinates. WACCM-D and MLS clearly produce similar structural patterns. Daytime peak in the mesosphere is stronger in MLS, potential reasons including model deficiencies in chemistry, solar radiation, and meridional circulation from gravity waves (Millán et al., 2015). Nighttime differences are smaller than during day. Overall, the model and observations agree reasonably well so that EPP detection possibilities should be similar, at least in terms of

background level of $HO_x$.

## 4.2   SPE thresholds

Figure 8 (left column) shows all detected nighttime SPE threshold values. As discussed in Sect. 3, correlations $\geq 0.35$ between the SPE indicator and $HO_x$ data are taken as indication of a reasonable threshold. These correlations are shown in Fig. 8 (right column). At higher latitudes in both hemispheres, i.e. at $60°$–$80°$, the correlations are distinctly high while

at other latitudes there is essentially no correlation. In the polar regions, the nightly threshold values are typically 100–175 protons $cm^{-2}s^{-1}sr^{-1}$. At magnetic latitude $80°$S the thresholds are lower than elsewhere, i.e. 75–110 protons $cm^{-2}s^{-1}sr^{-1}$, which would make this latitude band the best for SPE detection. The lower overall background concentration, especially in the SH due to the geomagnetic latitude distribution, is making detection easier, which leads to these lower thresholds. There are however larger variations at $80°$S as well, due to smaller amount of data available, causing weaker correlations in the MLS

observations. Overall, WACCM-D and MLS show similarly distributed threshold values. Comparing the thresholds from OH, SPEs are detected lower into the atmosphere in WACCM-D than MLS data, i.e. down to about 35 km and 50 km, respectively. This is seen in both hemispheres, in both thresholds and correlations, and can be explained by the MLS OH data becoming noisier in the stratosphere. In WACCM-D, the thresholds from OH and $HO_2$ are very consistent, although correlations are weaker in $HO_2$ in the Southern Hemisphere above 1 hPa.



Daytime SPE thresholds and correlations are shown in Fig. 9. Again, regions of high correlations are at high latitudes $60°$–$80°$ as expected. Because the daytime background concentrations of $HO_x$ are higher at most altitudes, the thresholds can be detected in a more restricted altitude range compared to nighttime. In WACCM-D, the detection can be done at 50–80 km. Both the threshold values and the correlations are very similar for $HO_2$ and OH data, even more so than at nighttime. For MLS

OH, the correlations are overall lower than in WACCM-D and there is a larger disparity between NH and SH. No thresholds are detected in SH, and in NH a total of four grid points, all at latitude $80°$N, have high enough correlations to qualify as reasonable thresholds. The WACCM-D thresholds vary mostly from 130 to 190 protons $cm^{-2}s^{-1}sr^{-1}$, though there are many values up to 275 protons $cm^{-2}s^{-1}sr^{-1}$, while MLS OH thresholds range from 140 to 180 protons $cm^{-2}s^{-1}sr^{-1}$. Some WACCM-D threshold values are above 300 protons $cm^{-2}s^{-1}sr^{-1}$. Thus, the daytime threshold values are higher than for the nighttime

and there is larger range of values as well.

We do not include MLS $HO_2$ in Figures 8–9, because we could not detect thresholds due to the noisiness of the data. However, the correlations between MLS $HO_2$ data and SPE indicators for both daytime and nighttime are shown in Figure 10. The correlations are quite uniformly around zero, although there are some larger values seen in the NH high latitudes and altitudes. Nevertheless, no effect of proton precipitation is detectable in the MLS $HO_2$ data. In the threshold detection analysis, the MLS

averaging kernels were not applied to the WACCM-D data. For comparison, WACCM-D $HO_2$ nighttime SPE thresholds with and without the MLS averaging kernel are shown in Fig. 11. Clearly, applying the averaging kernels leads to lower correlations and higher thresholds especially at altitudes above 1 hPa. However, thresholds can still be detected using WACCM-D data, especially in the NH.

In WACCM-D, the SPE forcing is applied uniformly at geomagnetic latitudes $> 60°$, and in $HO_x$ the impact is detected

at all the same latitudes. Observations also confirm this: the same latitude extent is seen in MLS OH observations at night. Verronen et al. (2007) used MLS OH data to study the latitudinal extent of the the proton forcing during the January 2005 SPE, comparing the results to a 1-D atmospheric simulation. They found that the lowest geomagnetic latitude affected by the SPE varied between $57°$ and $64°$ during the event. This agrees with our results with SPE impact detected at latitude bins poleward of $55°$, NH and SH. On the other hand, Heino et al. (2019) compared the latitudinal extent of 62 SPEs using cosmic radio

noise absorption from chain of riometer (relative ionospheric opacity meter) observations to those calculated from WACCM-D ionospheric output. They concluded that WACCM-D tends to overestimate the SPE impact at geomagnetic latitudes $> 66°$. However, Heino et al. (2019) included a large number of smaller events which are expected to affect the highest latitudes only (e.g. Rodger et al., 2006). For the set of events considered by us, the SPE impact seems to cover all geomagnetic latitudes above $60°$.

The SPE threshold fluxes from our $HO_x$ analysis are of the order of 100 protons $cm^{-2}s^{-1}sr^{-1}$. Thus our results are in agreement with a recent simulation study of SPE-driven atmospheric impacts which suggested little effect from SPEs with a smaller peak flux (Kalakoski et al., 2020). This detection limit means that of the 130 SPEs recorded in 2004–2018, 36 (28%) have a peak daily flux large enough for $HO_x$-based atmospheric detection.



### 4.3 RBE detection

The threshold and correlations for nightime RBE detection are shown in Fig. 12. In general, the $HO_x$ reaction to radiation belt electron precipitation is more limited in altitude and latitude compared to proton precipitation, which can be expected from spatial extent and energy range of observed electron fluxes.

Overall, we detect RBE only at latitudes poleward of $60°$ (NH and SH) and at altitudes from roughly 65 to 75 $km$. In WACCM-D, the detection threshold is mostly 0.85–1.35 $\times 10^4$ electrons $cm^{-2}s^{-1}sr^{-1}$, with lower values from WACCM-D when using the $HO_2$ data. With MLS OH data, in NH the detected RBE threshold values are lower than with WACCM-D OH, i.e. 0.85–1.25 $\times 10^4$ electrons $cm^{-2}s^{-1}sr^{-1}$, and cover a wider altitude range from 60 to 80 $km$. MLS OH seems to show a wider latitude range for detection in the correlations as well, extending over latitudes $50°–70°$, although RBE can be

detected only in one grid point outside $60°N$. No thresholds can be found for MLS $HO_2$ nighttime data, i.e. the situation is the same as for the SPE detection. There are no clear correlations between MLS $HO_2$ and RBE indicator (not shown). In daytime, all correlations between the RBE indicator and $HO_x$ concentrations are low. The number of detected daytime RBE indicator threshold values is only three, all with MLS OH (Fig. 13). These thresholds are 1.09–1.36 $\times 10^4$ electrons $cm^{-2}s^{-1}sr^{-1}$, at magnetic latitudes $60°–70°N$ and altitudes 0.01–0.0215 $hPa$.

In an attempt to improve some of the results, five-day averaged data were also examined. A moving five-day average was calculated from the $HO_x$ data and analysed as above with the SPE and RBE indicators. This was done to remove some of the noise, especially in the MLS $HO_2$, but the results were not improved. The data smoothing effectively flattened out the daily concentration peaks caused by events, which generally led to slightly lower correlations with the SPE and RBE indicators. Thus the daily values can be considered an optimal choice for EPP detection, taking into account that a typical SPE/RBE event

duration is days.

Although RBE forcing in WACCM-D is applied at geomagnetic latitudes $40°–72°$, the detection in $HO_x$ impact is only seen at $55°–65°$. Unlike the SPE forcing, RBE is not uniform over the latitude range but peaks at the heart of the outer radiation belt (van de Kamp et al., 2016). Thus, only this region which can be used to detect $HO_x$ impact. In the MLS data, i.e. in the correlation shown in Fig. 12, the RBE extent in the NH seems to reach into neighboring bins outside $55°–65°$ although the

correlation limit is not exceeded. This could indicate an underestimation in WACCM-D RBE forcing which is driven by the geomagnectic $Ap$ index.

The RBE threshold fluxes are of the order of $10^4$ electrons $cm^{-2}s^{-1}sr^{-1}$, i.e. 100 times larger fluxes than for SPEs. This is consistent considering that 100 $keV$ electrons ionize about 100 times less molecules than 10 $MeV$ protons while penetrating to about the same atmospheric altitude. Considering the time period 2004–2018, there are 192 days, i.e. 3.9% of all days,

which have RBE flux larger than the $10^4$ threshold. Note that our RBE thresholds are an order of magnitude larger than those given by Verronen et al. (2011). Analyzing four large RBE events, they estimated that it is not possible to detect $HO_x$ impact from electron forcing less than 10–30 $counts/s$ as measured by MEPED, and this count rate corresponds to fluxes of $1 - 3 \times 10^3$ electrons $cm^{-2}s^{-1}sr^{-1}$ (Evans and Greer, 2004).





# 5 Conclusions

In this study we have used atmospheric $HO_x$ data as a detector of EPP impact in the mesosphere and stratosphere. In a sense, WACCM-D simulations have provided us with the theoretical thresholds for the detection, while MLS observations are the present reality that is affected also by the quality of the measurements.

Overall, SPE impacts can be well detected using average nighttime OH data from MLS. Based on the WACCM-D results, detection should be possible also at daytime and using $HO_2$ data. In practise, however, the current MLS data do not have good enough signal-to-noise ratio to do this. RBE detection is possible as well, but only at nighttime and for more limited altitude and latitude ranges. Again, only MLS OH observations can be used for confident detection on a day-by-day basis.

Using the GOES $>10\,\mathrm{MeV}$ proton fluxes and POES 100–300 keV electron fluxes ($\mathrm{cm^{-2}s^{-1}sr^{-1}}$) as SPE and RBE proxies,

we find that the thresholds of the order of magnitude $10^2$ and $10^4$, respectively, have to be exceeded to cause detectable $HO_x$ impact. This limits the data usability to relatively large events. Note, however, that this does not mean that EPP with smaller fluxes is insignificant for the atmosphere. If applied for longer periods of time, EPP below the threshold limit can cause cumulative impacts on chemically long-lived species like $NO_x$.

Although the MLS $HO_2$ data were found to be too noisy for day-to-day EPP detection, they still have a great potential for

other purposes. For example, studies of solar-cycle variability in the mesosphere could greatly benefit from the long timeseries. Also, it has been shown, e.g. by Jackman et al. (2014), that the MLS $HO_2$ data are useful when largest solar proton events are studied.

*Code and data availability.* MLS data are available from the NASA Goddard Space Flight Center Earth Sciences (GES) Data and Information Services Center (DISC, https://mls.jpl.nasa.gov/data). All model data used are available from corresponding author by request. CESM

source code is distributed through a public subversion code repository (http://www.cesm.ucar.edu/models/cesm1.0/)

*Competing interests.* Authors declare that no competing interests are present.

*Acknowledgements.* The authors would like to thank the CHAMOS group (http://chamos.fmi.fi) for useful discussions.





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





**Figures**

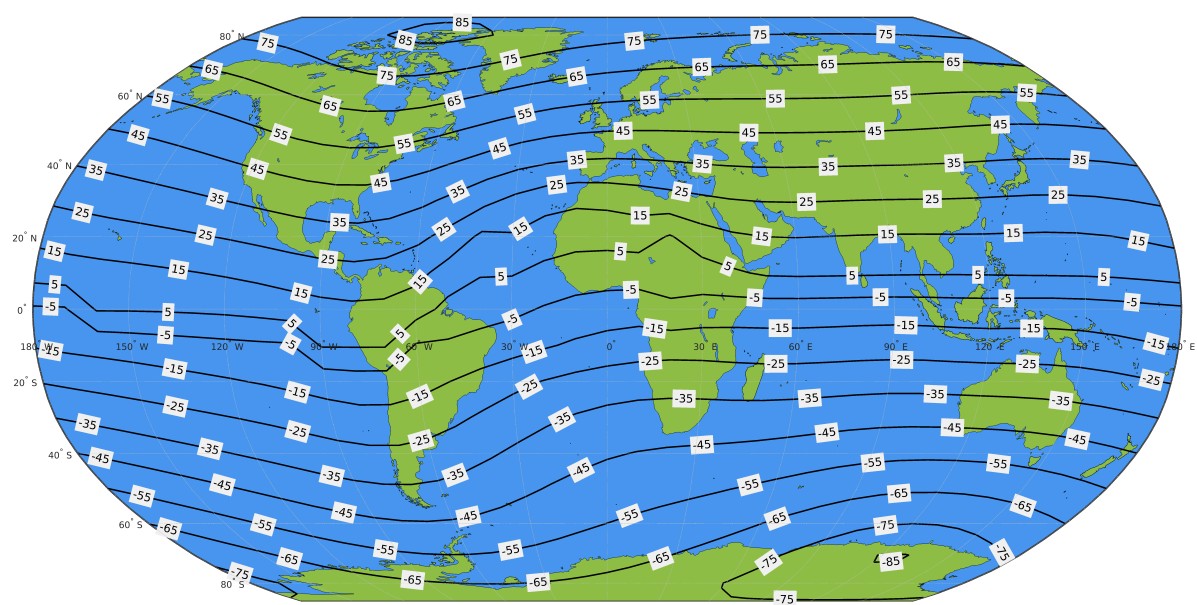

**Figure 1.** Geomagnetic latitude bin limits used in the analysis.



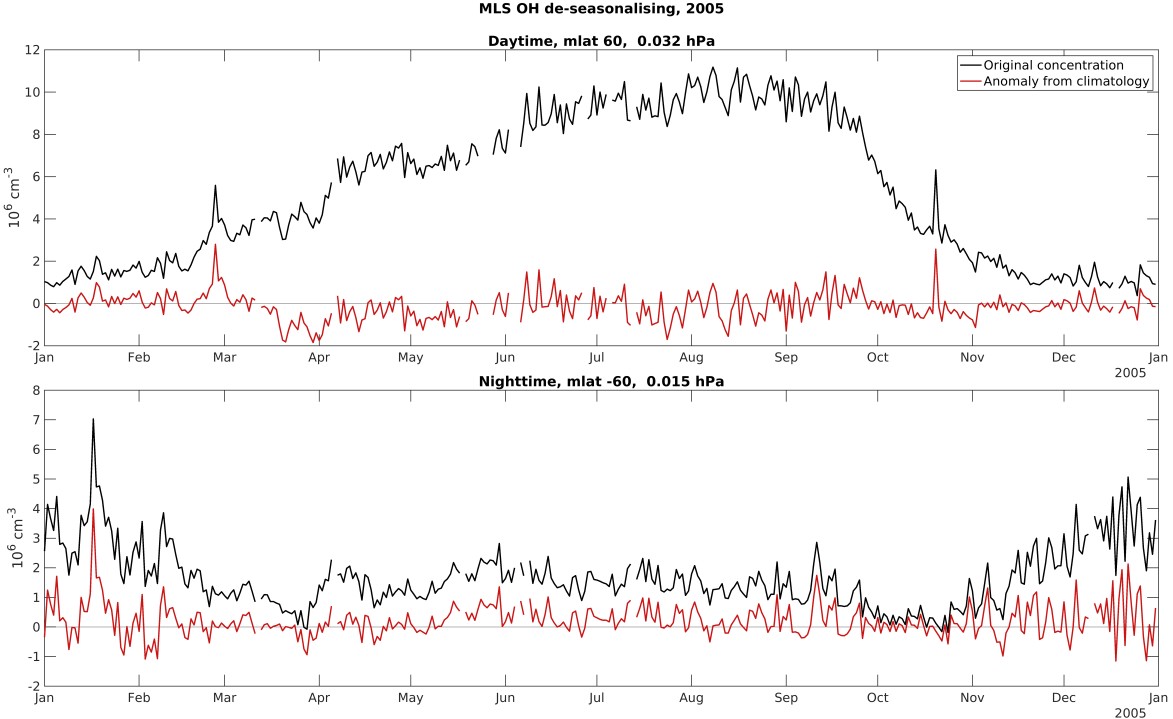

**Figure 2.** Daily MLS OH concentrations (in black) and the de-seasonalised anomalies from the climatology (red) in 2005. Daytime (top) on magnetic latitude $60°$N at $0.032$ hPa and nighttime (bottom) on magnetic latitude $60°$S at $0.015$ hPa.





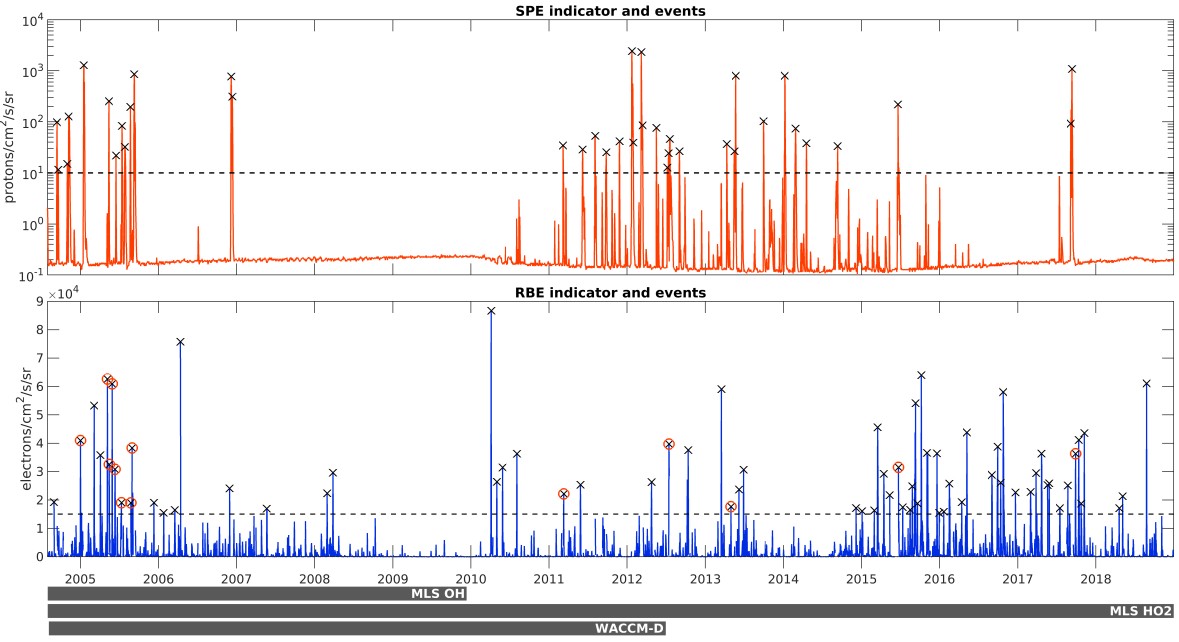

**Figure 3.** SPE indicator (top) and RBE indicator (bottom). Used precipitation limits for events are shown as dashed lines, and precipitation events are indicated by crosses. Encircled crosses mark RBE events with an SPE event in the same month. Availability of MLS OH and $HO_2$ as well as WACCM-D data used in the analysis is also shown below the graphs.



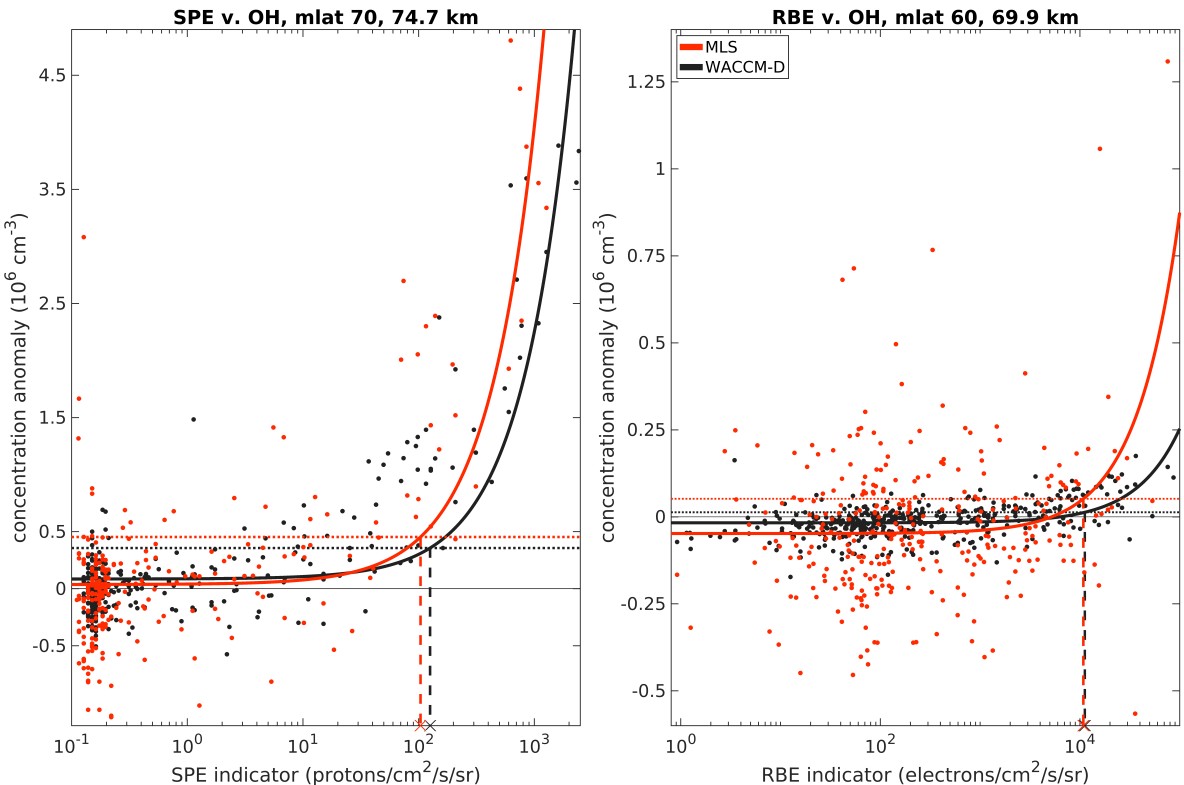

**Figure 4.** Precipitation threshold determination using nighttime OH, (left) SPE indicator at magnetic latitude 70°N, 0.0215 hPa (74.7 km) and (right) RBE indicator at magnetic latitude 60°N, 0.0464 hPa (69.9 km). MLS data in red and WACCM-D in black. Daily indicator and OH concentration anomaly from climatology value pairs are shown as dots, and the solid line shows the linear fit. Used limits (median + 0.5 × std) (dotted horizontal lines) and detected threshold values (dashed vertical lines) are also shown.





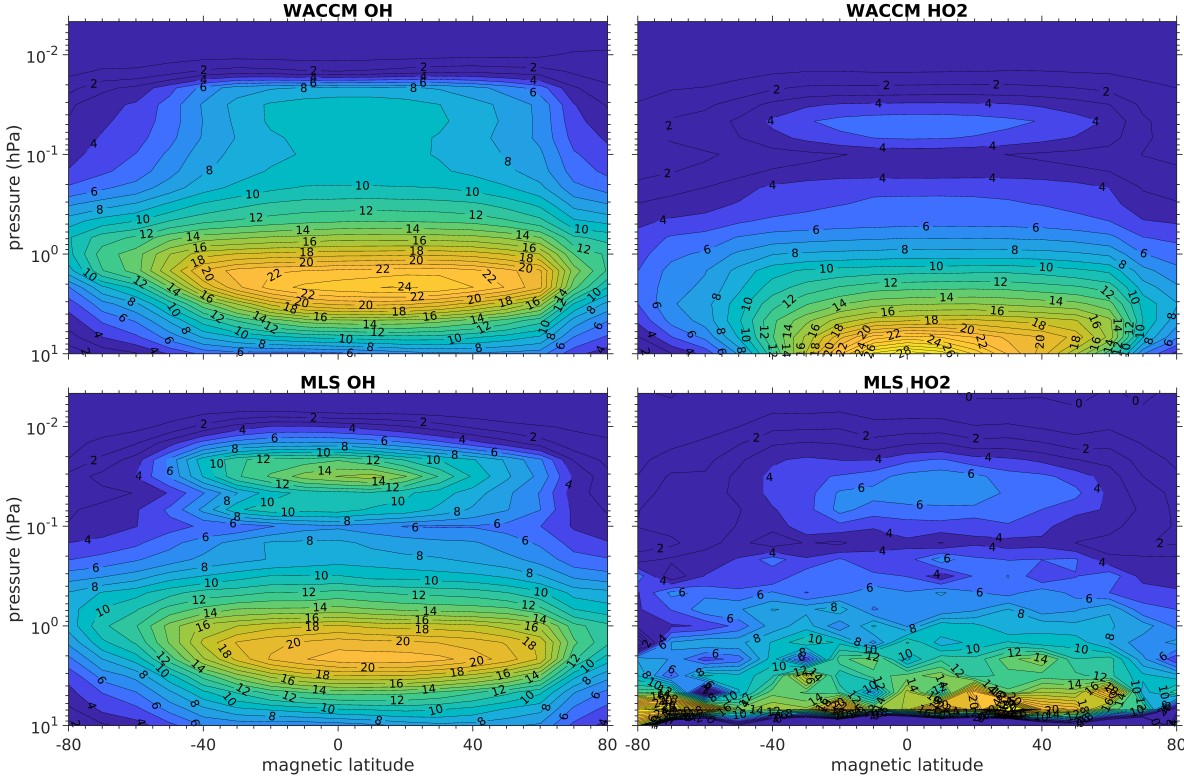

**Figure 5.** Monthly zonal averages of daytime concentrations in April 2009 from WACCM-D (top) and MLS (bottom). OH on the left, HO$_2$ on the right. Concentrations are in units $10^6$ cm$^{-3}$.





**Figure 6.** Monthly average daytime OH and $HO_2$ concentrations on magnetic latitude 70°N. From top to bottom: WACCM-D OH; MLS OH; WACCM-D $HO_2$; MLS $HO_2$. Concentrations are in units $10^6$ cm$^{-3}$.



**Figure 7.** Monthly average nighttime OH and HO$_2$ concentrations on magnetic latitude $60°$S. From top to bottom: WACCM-D OH; MLS OH; WACCM-D HO$_2$; MLS HO$_2$. Concentrations are in units $10^6$ cm$^{-3}$.



**Figure 8.** Nighttime SPE indicator thresholds (left) and corresponding correlations (right). WACCM-D HO$_2$ (top), WACCM-D OH (middle), and MLS OH (bottom).





**Figure 9.** Daytime SPE indicator thresholds (left) and corresponding correlations (right). WACCM-D HO$_2$ (top), WACCM-D OH (middle), and MLS OH (bottom).

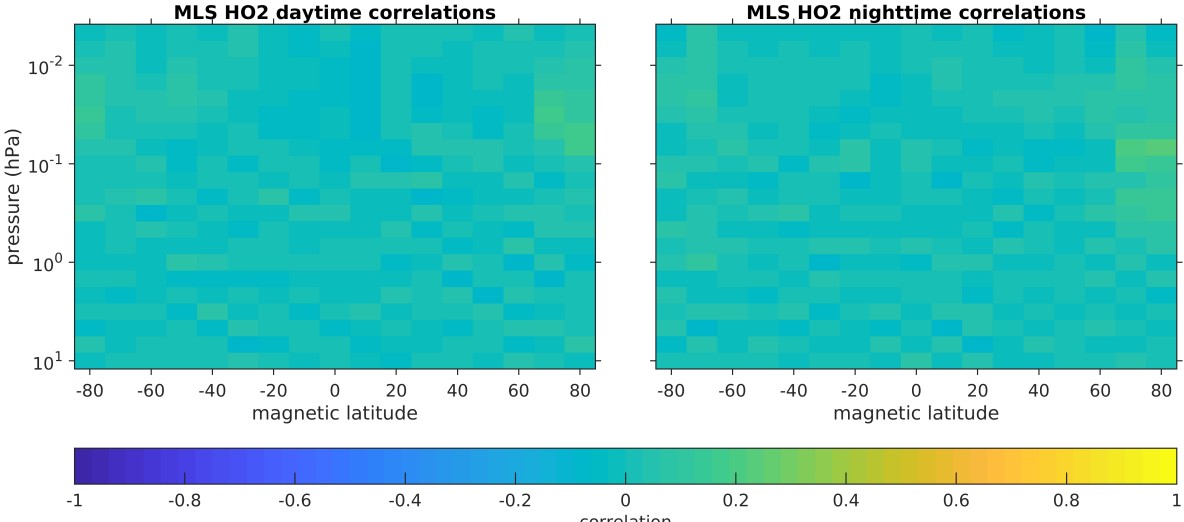

**Figure 10.** Correlations between SPE indicator and MLS $HO_2$ concentrations, (left) daytime and (right) nighttime. Due to the low correlations no threshold values could be detected.





**Figure 11.** Detected nighttime SPE thresholds (left) and corresponding correlations (right) for WACCM-D HO$_2$ without the use of averaging kernels (top), and with the MLS HO$_2$ averaging kernels (bottom). The top panels are the same as in Fig. 8.







**Figure 12.** Nighttime RBE indicator thresholds (left) and corresponding correlations (right). WACCM-D HO2 (top), WACCM-D OH (middle), and MLS OH (bottom).





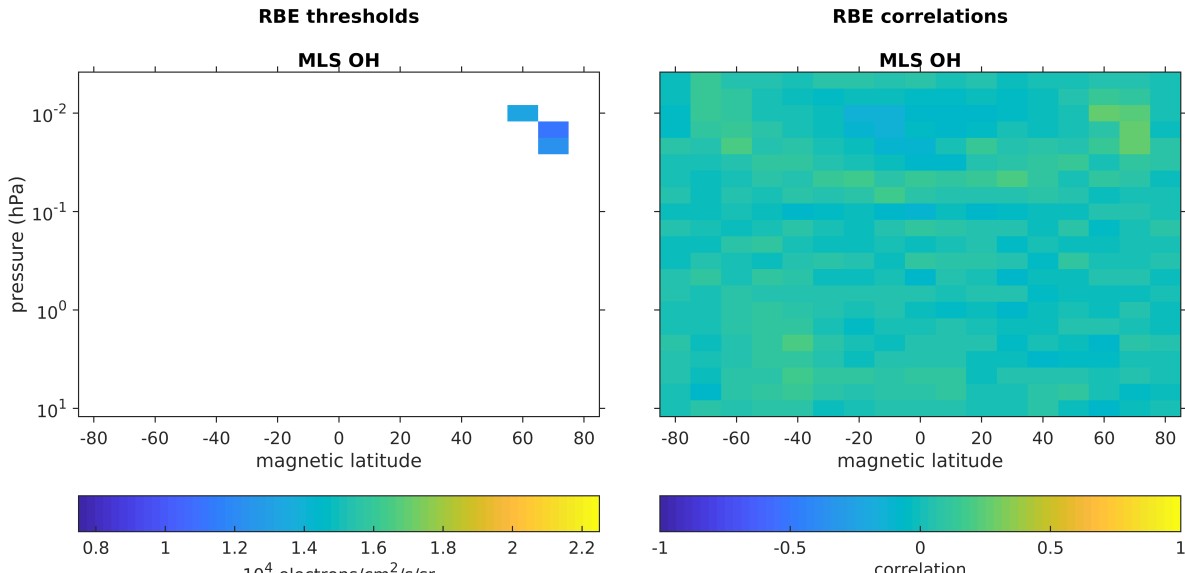

**Figure 13.** Daytime RBE indicator thresholds (left) and corresponding correlations (right) for MLS OH.



**Table 1.** Months included in the analysis, with dates of SPE and RBE flux peaks within each month also given. For peaks within five days of each other, only the date of the strongest is included. Dates marked by * indicate dates in months, where the event peak was in another month. RBE dates marked by × are not included in the analysis, due to SPE peak in the same month.

| month | SPE dates | RBE dates | month | SPE dates | RBE dates | month | SPE dates | RBE dates |
|---|---|---|---|---|---|---|---|---|
| 08/2004 | — | 31.08.2004 | 08/2011 | 05.08.2011 | — | 07/2015 | — | 13.07.2015 |
| 09/2004 | 14.09.2004 | — | 09/2011 | 26.09.2011 | — | 08/2015 | — | 17.08.2015 |
| | 20.09.2004 | | 11/2011 | 27.11.2011 | — | | | 27.08.2015 |
| 11/2004 | 01.11.2004 | — | 01/2012 | 24.01.2012 | — | 09/2015 | — | 11.09.2015 |
| | 08.11.2004 | | | 30.01.2012 | | | | 20.09.2015 |
| 01/2005 | 17.01.2005 | 02.01.2005× | 03/2012 | 08.03.2012 | — | 10/2015 | — | 08.10.2015 |
| 03/2005 | — | 07.03.2005 | | 14.03.2012 | | 11/2015 | — | 04.11.2015 |
| 04/2005 | — | 05.04.2005 | 04/2012 | — | 25.04.2012 | 12/2015 | — | 20.12.2015 |
| 05/2005 | 15.05.2005 | 08.05.2005× | 05/2012 | 17.05.2012 | — | | | 31.12.2015 |
| | | 16.05.2005× | 07/2012 | 07.07.2012 | 15.07.2012× | 01/2016 | — | 21.01.2016 |
| | | 30.05.2005× | | 13.07.2012 | | 02/2016 | — | 16.02.2016 |
| 06/2005 | 17.06.2005 | 12.06.2005× | | 19.07.2012 | | 04/2016 | — | 14.04.2016 |
| 07/2005 | 15.07.2005 | 12.07.2005× | 09/2012 | 02.09.2012 | — | 05/2016 | — | 08.05.2016 |
| | 29.07.2005 | | 10/2012 | — | 13.10.2012 | 09/2016 | — | 02.09.2016 |
| 08/2005 | 23.08.2005 | 25.08.2005× | 03/2013 | — | 17.03.2013 | | | 29.09.2016 |
| | | 31.08.2005× | 04/2013 | 11.04.2013 | — | 10/2016 | — | 13.10.2016 |
| 09/2005 | 10.09.2005 | — | 05/2013 | 17.05.2013 | 01.05.2013× | | | 25.10.2016 |
| 12/2005 | — | 11.12.2005 | | 23.05.2013 | | 12/2016 | — | 21.12.2016 |
| 01/2006 | — | 26.01.2006 | 06/2013 | — | 07.06.2013 | 03/2017 | — | 02.03.2017 |
| 03/2006 | — | 19.03.2006 | | | 29.06.2013 | | | 27.03.2017 |
| 04/2006 | — | 14.04/2006 | 09/2013 | 30.09.2013* | — | 04/2017 | — | 22.04.2017 |
| 11/2006 | — | 30.11.2006 | 10/2013 | 01.10.2013 | — | 05/2017 | — | 20.05.2017 |
| 12/2006 | 07.12.2006 | — | 01/2014 | 08.01.2014 | — | | | 28.05.2017 |
| | 13.12.2006 | | 02/2014 | 27.02.2014 | — | 07/2017 | — | 17.07/2017 |
| 05/2007 | — | 23.05.2007 | 03/2014 | 01.03.2014* | — | 08/2017 | — | 23.08.2017 |
| 02/2008 | — | 29.02.2008 | 04/2014 | 19.04.2014 | — | 09/2017 | 05.09.2017 | 28.09.2017× |
| 03/2008 | — | 27.03.2008 | 09/2014 | 12.09.2014 | — | | 11.09.2017 | |
| 04/2010 | — | 06.04.2010 | 12/2014 | — | 07.12.2014 | 10/2017 | — | 13.10.2017 |
| 05/2010 | — | 02.05.2010 | 01/2015 | — | 04.01.2015 | | | 24.10.2017 |
| | | 29.05.2010 | 03/2015 | — | 02.03.2015 | 11/2017 | — | 08.11.2017 |
| | | | | | 18.03.2015 | 04/2018 | — | 20.04.2018 |
| 08/2010 | — | 04.08.2010 | | | 16.04.2015 | 05/2018 | — | 06.05.2018 |
| 03/2011 | 08.03.2011 | 11.03.2011× | 04/2015 | — | 13.05.2015 | 08/2018 | — | 26.08.2018 |
| 05/2011 | — | 28.05.2011 | 05/2015 | — | | | | |
| 06/2011 | 07.06.2011 | — | 06/2015 | 21.06.2015 | 23.06.2015× | | | |