# Peer review of "Odd hydrogen response thresholds for indication of solar proton and electron impact in the mesosphere and stratosphere"

_Annales Geophysicae, 2020_

## Referee Comment (RC1) · Anonymous Referee #1 · 26 Jun 2020

General comments:

In this paper, the response of OH and HO2 in the stratosphere and mesosphere to large particle precipitation events – solar proton events and electron precipitation events – is investigated based on observations from the MLS satellite and model results from the WACCM chemistry-climate model. In particular, increases in both data-sets during periods of increased proton or electron flux are used to determine a threshold flux above which an observable response can be expected. The topic is of great interest as OH and HO2 observations during such particle precipitation events are good indicators of an atmospheric impact, and of potential great use to evaluate the particle impact in

chemistry-climate models used to study the climate feedback of these events. The paper is also very well written. However, in my opinion there is a problem with the methodology used to calculate the threshold which potentially leads to a high bias. I have summarized my concern below (specific comments), and am looking forward to a productive discussion of this point.

Specific comments:

Page 5, lines 20 and following, discussion of threshold determination: I have two comments on the determination of the threshold, which in my opinion could be improved considerably.

- Line 23: you use a linear fit between two datasets which have a very different range of variability: the particle fluxes vary over nearly six orders of magnitude, the anomalies vary by less than a factor of ten. I think this cannot work. Everything lower than about 10% of the maximum value of the x vector (particle flux) will be interpreted as essentially zero by the fitting routine, so whatever threshold value you derive here is probably within the uncertainty of the fit. You would see this clearly if you plotted the values on a linear scale – you lose the information about the lower flux values if you plot the flux on a linear, and not on a logarithmic scale, and the same is true if you do a linear fit on these values. You can see quite clearly in the left panel of Figure 4 that the fit did not work – just look at the black dots and black line (WACCM-D data and fit): for fluxes between 10 and 100, y-values are still rather high, but the dots are mostly to the left of the fitting line, that is, the lower flux values are not well represented by the fit. Only values with flux (SPE indicator) values above about 100 are well represented by the fit, as only those can be considered if the SPE indicator is used as a linear parameter. This also means that you overestimate the threshold value, and I think that this has to happen: the linear fit provides an artificial upper limit of about 10% of the highest value. If you just look at the black dots – there are a lot of dots between SPE flux values of 10 and 100 which are significantly above the fitting curve. If you just look at these dots, the threshold is probably around 10, not larger than 100 as your fit
Interactive
comment

provides. I think using the log of the flux (SPE/RBE indicator) and a non-linear fitting function (polynomial) will provide a much better fit which also can account for the low flux values. If you do this with a multi-linear regression algorithm, you can still use the correlation coefficient as a measure of fitting quality.

- I think using half the standard deviation as a measure of significant enhancement is too low – this would be still in the noise floor. When you do the fit against log(flux) as suggested above, you can probably afford to use one standard deviation, and still get a lower threshold.

Technical corrections:

Page 4, lines 19-20: I think it would be more consistent to use the WACCM density for conversion, not the MLS $HO_2$ density. Because even if the output format for WACCM species may be mixing ratio, internally number densities are likely used for the calculation of photochemistry.

Page 4, line 21: note formatting of $HO_2$

Page 5, line 18-20: but would you not expect a difference in the HOx loss rates between summer and winter which likely affect the observed increase?

Page 6, line 30: missing full stop after "levels"

---

## Referee Comment (RC2) · Anonymous Referee #2 · 2 Jul 2020

The paper focuses on the impact of energetic particle precipitation (EPP, both proton and electron events) on the polar middle atmosphere. The overall goal is to determine the EPP flux thresholds at various altitudes and locations by using odd hydrogen species observed by MLS/Aura satellite and simulated by WACCM-D. Due to the significant uncertainties in satellite measurements of energetic particles, especially for electron precipitation, this study is useful. Although the study does not present critical advances, the paper is well written, the methodology is sound, and the results are in line with previous studies. Overall, I suggest publication subjected to address the following comments.

[Figure]

Please try to make a more focused paper. Thirteen figures are too many for this study and make the reading difficult. It would be beneficial to try combining some of them. For example, three figures for showing the comparison of the climatology between MLS and WACCM are excessive. Other figures (at least Fig. 2 and Fig. 11) could be removed or included as supplementary information.

Abstract: I would like to see more emphasis on your new results. The current abstract mainly describes the state-of-the-art of this topic. A more in-depth discussion on the difference between observations and simulations at lower altitudes could be useful here as well as in the main text. The use of the improved MLS HO2 dataset should be mentioned in the abstract.

Methods: Why did you not simply compute the HOx anomalies with respect to the previous days? The EPP-induced HOx enhancements are spikes lasting a few days at most. On the other hand, there is a consistent interannual variability, especially in the northern polar vortex. For example, how did you deal with the SSW occurrence? SSWs can affect the nighttime mesospheric OH layer for weeks (see Winick et al., 2009, Damiani et al, 2010). I think this issue should be at least mentioned. I know that it is common using the flux at energies >10 MeV, but what about the higher or lower energies? For example, I could expect a better correlation in the upper (lower) mesosphere with energies >5 (30) MeV.

Figure 10: By using the standard MLS HO2 dataset, Jackman et al. 2011 and 2014 showed evident MLS HO2 variability during the SPEs of January 2005 and January and March 2012. Here, it is puzzling that the nighttime northern hemisphere did not show any evident HO2 change (i.e., correlation < 0.35). If so, why did you use the new HO2 dataset of MillaÌAn et al. (2015)? What's the advantage of using this dataset for EPP-related studies? It could be good showing some comparison between the standard and the new HO2 dataset. If it is not possible to reproduce Figure 10 with the standard dataset, I suggest including at least a case study for a single event.

Pag. 7, l30-34: In Fig. 8, you showed SPE-related changes down to about 35 km in WACCM-D and 50 km in MLS observations. This point deserves more discussion because the evaluation of the direct particle impact at these altitudes is very important. You correctly highlighted the issue of the MLS observations i.e., OH data become noisier in the stratosphere. Therefore, you cannot accurately evaluate the SPE thresholds in this region by using MLS $HO_x$. Nevertheless, you could potentially do it with other MLS products (e.g., chlorine species) which are properly simulated by WACCM (Funke et al., 2011).

Pag. 7, l30-34: Why? Some issues with the data sampling? Perhaps SPE-related effects could be highlighted even better by using geographic latitudes.

References: Funke, B., et al., Composition changes after the "Halloween" solar proton event: the High-Energy Particle Precipitation in the Atmosphere (HEPPA) model versus MIPAS data intercomparison study, Atmos. Chem. Phys., 11, 9089–9139, https://doi.org/10.5194/acp- 11-9089-2011, 2011.

Damiani A., Storini M., Santee M. L., Wang S., Variability of the nighttime OH layer and mesospheric ozone at high latitudes during Northern winter: Influence of meteorology, Atmospheric Chemistry and Physics, 10, 10291–10303, 2010.

Winick, J. R., Wintersteiner, P. P., Picard, R. H., Esplin, D., Mlynczak, M. G., Russell III, J. M., and Gordley, L. L.: OH layer characteristics during unusual boreal winters of 2004 and 2006, J. Geophys. Res., 114, A02303, doi:10.1029/2008JA013688, 2009.

---

## Author Comment (AC2) · 18 Sep 2020

Please find below our answers (in blue) to the comments (in black).

**Response to the comments of Referee #2**

The paper focuses on the impact of energetic particle precipitation (EPP, both proton

and electron events) on the polar middle atmosphere. The overall goal is to determine the EPP flux thresholds at various altitudes and locations by using odd hydrogen species observed by MLS/Aura satellite and simulated by WACCM-D. Due to the significant uncertainties in satellite measurements of energetic particles, especially for electron precipitation, this study is useful. Although the study does not present critical advances, the paper is well written, the methodology is sound, and the results are in line with previous studies. Overall, I suggest publication subjected to address the following comments.

Response to the general comments: We thank the reviewer for the constructive comments. We also appreciate the time devoted to the evaluation of our paper.

Please try to make a more focused paper. Thirteen figures are too many for this study and make the reading difficult. It would be beneficial to try combining some of them. For example, three figures for showing the comparison of the climatology between MLS and WACCM are excessive. Other figures (at least Fig. 2 and Fig. 11) could be removed or included as supplementary information.

We have removed Figures 5 and 11 to make paper more focused. Figure 2 we kept, because it is needed to demonstrate our methods. Figures 6–7 were also kept, as we would like to show comparisons of both day and night $HO_x$. The text has been revised to accommodate the changes in the figures.

Abstract: I would like to see more emphasis on your new results. The current abstract

mainly describes the state-of-the-art of this topic. A more in-depth discussion on the difference between observations and simulations at lower altitudes could be useful here as well as in the main text. The use of the improved MLS HO2 dataset should be mentioned in the abstract.

We agree that the abstract should have more details on the results and numeric information can be increased. We have thus revised the abstract in these aspects

Methods: Why did you not simply compute the HOx anomalies with respect to the previous days? The EPP-induced HOx enhancements are spikes lasting a few days at most. On the other hand, there is a consistent interannual variability, especially in the northern polar vortex. For example, how did you deal with the SSW occurrence? SSWs can affect the nighttime mesospheric OH layer for weeks (see Winick et al.,2009, Damiani et al, 2010). I think this issue should be at least mentioned. I know that it is common using the flux at energies >10 MeV, but what about the higher or lower energies? For example, I could expect a better correlation in the upper (lower)mesosphere with energies >5 (30) MeV.

We use a daily climatology in our analysis to calculate $HO_x$ concentration anomalies in order to combine events from different seasons, as our aim is to define general threshold values for EPP detection in $HO_x$. We agree that this method may not be an optimal solution for some of the events, but argue that as a whole it is still appropriate. Day-to-day variability from sources other than EPP also exist in the data, and we believe the use of previous days would be more likely to include this type of uncertainty in the analysis.

We did not take SSWs into account in the analysis, so this will add to the background variability. However, we show the results separately for the SH and NH, and in our analysis the SH data do not have SSWs. Although the SSWs related variability is included in our analysis, the results are largely consistent between SH and NH, which indicates that they are robust despite background dynamical differences. We have added discussion on this to the revised manuscript.

We do not expect significant differences to arise from the use of different proton flux energies, because the measurements from the GOES proton energy channels correlate well with each other. For electrons we are not aware of a good alternative for the indicator used in our analysis, while we note that the current one could possibly focus our capability to middle mesosphere detection only. We have added discussion on this to the revised manuscript.

Figure 10: By using the standard MLS HO2 dataset, Jackman et al. 2011 and 2014 showed evident MLS HO2 variability during the SPEs of January 2005 and January and March 2012. Here, it is puzzling that the nighttime northern hemisphere did not show any evident HO2 change (i.e., correlation < 0.35). If so, why did you use the new HO2 dataset of Millán et al. (2015)? What's the advantage of using this dataset for EPP-related studies? It could be good showing some comparison between the standard and the new HO2 dataset. If it is not possible to reproduce Figure 10 with the standard dataset, I suggest including at least a case study for a single event.

We use the offline $HO_2$ data because it offers better S/N ratio in general and an extended altitude range when compared to the standard MLS data.

We checked the standard and offline data for the January 2005, as well as January and March 2012 events. Compared to WACCM $HO_2$, which shows a clear and extended response to SPEs, both the standard and offline data show only patchy responses around 0.1 hPa at high latitudes. Overall, in this case the differences in results between the standard and offline data are small, but the offline data provides a better altitude extent. We also analysed our full set of SPEs using the standard data, and there are no thresholds detected.

We decided not to include any comparisons between the standard and offline $HO_2$ data in our paper, because we see that to be outside the scope of this study and would be a distraction from the main points and focus. However, we have revised the text and added justification for the use of the offline data to the Data and models section.

Pag. 7, l30-34: In Fig. 8, you showed SPE-related changes down to about 35 km in WACCM-D and 50 km in MLS observations. This point deserves more discussion because the evaluation of the direct particle impact at these altitudes is very important. You correctly highlighted the issue of the MLS observations i.e., OH data become noisier in the stratosphere. Therefore, you cannot accurately evaluate the SPE thresholds in this region by using MLS HOx. Nevertheless, you could potentially do it with other MLS products (e.g., chlorine species) which are properly simulated by WACCM (Funke et al., 2011).

We have highlighted this matter in our conclusions and also discuss the use of other species in the detection of EPPs in the revised manuscript. Kalakoski et al. (2020) showed SPE-related increases in WACCM-D $Cl_x$ down to 1 hPa lasting around a week following an event. They also showed a response in $HNO_3$ above 1 hPa with slightly

longer duration (10 days), and enhancements lasting 20–30 days following SPEs below the 1 hPa level.

Pag. 7, l30-34: Why? Some issues with the data sampling? Perhaps SPE-related effects could be highlighted even better by using geographic latitudes.

We are somewhat unsure what this refers to. There may be an error in the comment, since the lines given as reference are exactly the same as for the previous comment, and the comment does not seem relevant to these lines.

This comment may be in reference to p. 8, lines 4–7 discussing differences in daytime threshold detection between SH and NH. The differences do arise from sampling of the data and the distribution of the magnetic latitudes and the geographic latitude coverage of the MLS measurements (82S to 82N (Waters et al., 2006)). These together lead to the daytime magnetic latitude bin 80N having more than 35% more MLS measurement points than the 80S bin. This is likely to cause the discrepancy between NH and SH in daytime SPE detection. Hence this is an issue of data availability at very high latitudes but does not mean that geographic latitudes are necessarily better suited to the study of SPE effects. Using geographic latitudes would not increase the amount of observations available for analysis, the data would simply be binned in a different manner. We believe the geomagnetic latitudes are a better choice, especially where the latitudes are evenly covered by the measurements, because $HO_x$ impact is clearly seen along the geomagnetic latitudes (e.g. Andersson et al., 2014).

**References**

Andersson, M. E., Verronen, P. T., Rodger, C. J., Clilverd, M. A., and Wang, S.: Longitudinal hotspots in the mesospheric OH variations due to energetic electron precipitation, Atmos. Chem. Phys., 14, 1095–1105, https://doi.org/10.5194/acp-14-1095-2014, 2014.

Kalakoski, N., Verronen, P. T., Seppälä, A., Szeląg, M. E., Kero, A., and Marsh, D. R.: Statistical response of middle atmosphere composition to solar proton events in WACCM-D simulations: importance of lower ionospheric chemistry, Atmos. Chem. Phys., 20, 8923–8938, https://doi.org/10.5194/acp-20-8923-2020, 2020.

Waters, J. W., Froidevaux, L., Harwood, R. S., Jarnot, R. F., Pickett, H. M., Read, W. G., Siegel, P. H., Cofield, R. E., Filipiak, M. J., Flower, D. A., Holden, J. R., Lau, G. K., Livesey, N. J., Manney, G. L., Pumphrey, H. C., Santee, M. L., Wu, D. L., Cuddy, D. T., Lay, R. R., Loo, M. S., Perun, V. S., Schwartz, M. J., Stek, P. C., Thurstans, R. P., Boyles, M. A., Chandra, K. M., Chavez, M. C., Chen, G.-S., Chudasama, B. V., Dodge, R., Fuller, R. A., Girard, M. A., Jiang, J. H., Jiang, Y., Knosp, B. W., Labelle, R. C., Lam, J. C., Lee, A. K., Miller, D., Oswald, J. E., Patel, N. C., Pukala, D. M., Quintero, O., Scaff, D. M., Vansnyder, W., Tope, M. C., Wagner, P. A., and Walch, M. J.: The Earth Observing System Microwave Limb Sounder (EOS MLS) on the Aura satellite, IEEE Trans. Geosci. Remote Sens., 44, 1075–1092, https://doi.org/10.1109/TGRS.2006.873771, 2006.

---

## Author Response (AR1)

**Authors' response to referees' comments on "Odd hydrogen response thresholds for indication of solar proton and electron impact in the mesosphere and stratosphere" by Häkkilä et al.**

Please find below our answers (in blue) to the comments (in black).

**Response to the comments of Referee #1**

5   In this paper, the response of OH and HO2 in the stratosphere and mesosphere to large particle precipitation events – solar proton events and electron precipitation events – is investigated based on observations from the MLS satellite and model results from the WACCM chemistry-climate model. In particular, increases in both data-sets during periods of increased proton or electron flux are used to determine a threshold flux above which an observable response can be expected. The topic is of great interest as OH and HO2 observations during such particle precipitation events are good indicators of an

10  atmospheric impact, and of potential great use to evaluate the particle impact in chemistry-climate models used to study the climate feedback of these events. The paper is also very well written. However, in my opinion there is a problem with the methodology used to calculate the threshold which potentially leads to a high bias. I have summarized my concern below (specific comments), and am looking forward to a productive discussion of this point.

Response to general comments: We would like to thank the referee for his/her positive comments and appreciate the time
15  devoted to the evaluation of our paper.

Specific comments:
Page 5, lines 20 and following, discussion of threshold determination: I have two comments on the determination of the threshold, which in my opinion could be improved considerably.
- Line 23: you use a linear fit between two datasets which have a very different range of variability: the particle fluxes vary
20  over nearly six orders of magnitude, the anomalies vary by less than a factor of ten. I think this cannot work. Everything lower than about 10% of the maximum value of the x vector (particle flux) will be interpreted as essentially zero by the fitting routine, so whatever threshold value you derive here is probably within the uncertainty of the fit. You would see this clearly if you plotted the values on a linear scale – you lose the information about the lower flux values if you plot the flux on a linear, and not on a logarithmic scale, and the same is true if you do a linear fit on these values. You can see quite clearly in the left
25  panel of Figure 4 that the fit did not work – just look at the black dots and black line (WACCM-D data and fit):for fluxes between 10 and 100, y-values are still rather high, but the dots are mostly to the left of the fitting line, that is, the lower flux values are not well represented by the fit. Only values with flux (SPE indicator) values above about 100 are well represented by the fit, as only those can be considered if the SPE indicator is used as a linear parameter. This also means that you overestimate the threshold value, and I think that this has to happen: the linear fit provides an artificial upper limit of about
30  10% of the highest value. If you just look at the black dots – there are a lot of dots between SPE flux values of 10 and 100 which are significantly above the fitting curve. If you just look at these dots, the threshold is probably around 10, not larger than 100 as your fit provides. I think using the log of the flux (SPE/RBE indicator) and a non-linear fitting function (polynomial) will provide a much better fit which also can account for the low flux values. If you do this with a multi-linear regression algorithm, you can still use the correlation coefficient as a measure of fitting quality.

35  Thank you for the insightful comment. We have re-examined our threshold determination method. In addition to the suggested use of a logarithm of EPP flux values, we also tried our method using a square root of the EPP fluxes. The latter was inspired by the previous work of Verronen et al. (2011). As suggested, we tested a 2nd degree polynomial with the logarithm, and one standard deviation as limit of significant concentration enhancement (see also the next comment and response). We also used one standard deviation for the square root method, but used a first degree polynomial like Verronen et al. (2011).

As suggested by the reviewer, the threshold values are indeed lower for SPEs for both tested methods when compared to our original approach. For electron precipitation, however, the tested methods resulted in higher threshold values, likely due to the use of one standard deviation instead of only a half.

In our analysis, the two tested fits were of similar quality and better than our original method. For SPEs, the square root method produces a factor of $\approx 2$ larger threshold values than the logarithm method. After consideration, we have selected the results from the square root fitting method for the revised paper. This is mainly due to higher correlations produced by the square root method, leading to greater altitude-latitude extent of detected threshold values. This is somewhat to be expected, as the square root method uses a linear fit compared to the second degree polynomial used with the logarithm method. This difference in number of detected threshold values could therefore perhaps be accounted for by lowering the correlation limit, but we do not think there is much room to lower the current limit of 0.35 without losing in the screening effect. We also consider the chemistry-based reasoning behind the square root method to be in its favor (Verronen et al., 2011).

In the revised manuscript, Figures 4, 8–10, and 12–13 were updated (Figure 11 was removed as suggested by referee #2), and text was changed accordingly. Mostly this was changing the description of the method and the threshold values, since qualitatively our results and conclusions did not change.

- I think using half the standard deviation as a measure of significant enhancement is too low – this would be still in the noise floor. When you do the fit against log(flux) as suggested above, you can probably afford to use one standard deviation, and still get a lower threshold.

We changed the method to use one standard deviation. As stated above, the detected SPE thresholds (STD + SQRT fitting) are still lower than the originals. However the electron precipitation thresholds are higher with the revised method, at least partly due to the change from a half to full STD. Using half STD with the square root results in threshold values similar or slightly lower than originally, but one STD raises these thresholds to be mostly around 1.5 times greater than with the original method. Regardless we still choose the revised method with one STD as the limit, since the method works clearly better with SPEs.

Technical corrections:
Page 4, lines 19-20: I think it would be more consistent to use the WACCM density for conversion, not the MLS $HO_2$ density. Because even if the output format for WACCM species may be mixing ratio, internally number densities are likely used for the calculation of photochemistry.

Both WACCM and MLS provide mixing ratios, although we use concentrations in the fitting. Conversion is made consistently for both, using MLS-derived total densities. Since we are aiming at EPP detection and not chemistry in detail, we do not think this is a crucial factor. Differences in total densities between WACCM-D and MLS should be relatively small in the middle atmosphere compared to the maximum changes in $HO_x$ from EPP (from up to hundreds of percent to an order of magnitude).

Page 4, line 21: note formatting of HO2

We have corrected the formatting in the revised manuscript.

Page 5, line 18-20: but would you not expect a difference in the HOx loss rates between summer and winter which likely affect the observed increase?

We do not expect a large impact from the possible $HO_x$ loss differences between summer and winter. $HO_x$ loss is controlled mainly by $HO_x$ recombination and reactions with atomic oxygen (Canty and Minschwaner, 2002). The chemical lifetime of $HO_x$ is less than one day at altitudes below 80 km (Pickett et al., 2006). The larger impact will come through background

$HO_x$ production which has both seasonal and diurnal variability. However, our analysis takes this into account by using climatology-corrected anomalies and by considering day and night times separately.

Page 6, line 30: missing full stop after "levels"

We have corrected this in the revised manuscript.

5 **Response to the comments of Referee #2**

The paper focuses on the impact of energetic particle precipitation (EPP, both proton and electron events) on the polar middle atmosphere. The overall goal is to deter-mine the EPP flux thresholds at various altitudes and locations by using odd hydrogen species observed by MLS/Aura satellite and simulated by WACCM-D. Due to the significant uncertainties in satellite measurements of energetic particles, especially for electron precipitation, this study is useful. Although the study does not
10 present critical advances, the paper is well written, the methodology is sound, and the results are in line with previous studies. Overall, I suggest publication subjected to address the following comments.

Response to the general comments: We thank the reviewer for the constructive comments. We also appreciate the time devoted to the evaluation of our paper.

Please try to make a more focused paper. Thirteen figures are too many for this study and make the reading difficult. It would
15 be beneficial to try combining some of them. For example, three figures for showing the comparison of the climatology between MLS and WACCM are excessive. Other figures (at least Fig. 2 and Fig. 11) could be removed or included as supplementary information.

We have removed Figures 5 and 11 to make paper more focused. Figure 2 we kept, because it is needed to demonstrate our methods. Figures 6–7 were also kept, as we would like to show comparisons of both day and night $HO_x$. The text has been
20 revised to accommodate the changes in the figures.

Abstract: I would like to see more emphasis on your new results. The current abstract mainly describes the state-of-the-art of this topic. A more in-depth discussion on the difference between observations and simulations at lower altitudes could be useful here as well as in the main text. The use of the improved MLS HO2 dataset should be mentioned in the abstract.

We agree that the abstract should have more details on the results and numeric information can be increased. We have thus
25 revised the abstract in these aspects

Methods: Why did you not simply compute the HOx anomalies with respect to the previous days? The EPP-induced HOx enhancements are spikes lasting a few days at most. On the other hand, there is a consistent interannual variability, especially in the northern polar vortex. For example, how did you deal with the SSW occurrence? SSWs can affect the nighttime mesospheric OH layer for weeks (see Winick et al.,2009, Damiani et al, 2010). I think this issue should be at least mentioned.
30 I know that it is common using the flux at energies >10 MeV, but what about the higher or lower energies? For example, I could expect a better correlation in the upper (lower)mesosphere with energies >5 (30) MeV.

We use a daily climatology in our analysis to calculate $HO_x$ concentration anomalies in order to combine events from different seasons, as our aim is to define general threshold values for EPP detection in $HO_x$. We agree that this method may not be an optimal solution for some of the events, but argue that as a whole it is still appropriate. Day-to-day variability from
35 sources other than EPP also exist in the data, and we believe the use of previous days would be more likely to include this type of uncertainty in the analysis.

We did not take SSWs into account in the analysis, so this will add to the background variability. However, we show the results separately for the SH and NH, and in our analysis the SH data do not have SSWs. Although the SSWs related variability is included in our analysis, the results are largely consistent between SH and NH, which indicates that they are robust despite background dynamical differences. We have added discussion on this to the revised manuscript.

5  We do not expect significant differences to arise from the use of different proton flux energies, because the measurements from the GOES proton energy channels correlate well with each other. For electrons we are not aware of a good alternative for the indicator used in our analysis, while we note that the current one could possibly focus our capability to middle mesosphere detection only. We have added discussion on this to the revised manuscript.

Figure 10: By using the standard MLS HO2 dataset, Jackman et al. 2011 and 2014 showed evident MLS HO2 variability
10  during the SPEs of January 2005 and January and March 2012. Here, it is puzzling that the nighttime northern hemisphere did not show any evident HO2 change (i.e., correlation < 0.35). If so, why did you use the new HO2 dataset of Millán et al. (2015)? What's the advantage of using this dataset for EPP-related studies? It could be good showing some comparison between the standard and the new HO2 dataset. If it is not possible to reproduce Figure 10 with the standard dataset, I suggest including at least a case study for a single event.

15  We use the offline $HO_2$ data because it offers better S/N ratio in general and an extended altitude range when compared to the standard MLS data.

We checked the standard and offline data for the January 2005, as well as January and March 2012 events. Compared to WACCM $HO_2$, which shows a clear and extended response to SPEs, both the standard and offline data show only patchy responses around 0.1 hPa at high latitudes. Overall, in this case the differences in results between the standard and offline data
20  are small, but the offline data provides a better altitude extent. We also analysed our full set of SPEs using the standard data, and there are no thresholds detected.

We decided not to include any comparisons between the standard and offline $HO_2$ data in our paper, because we see that to be outside the scope of this study and would be a distraction from the main points and focus. However, we have revised the text and added justification for the use of the offline data to the Data and models section.

25  Pag. 7, l30-34: In Fig. 8, you showed SPE-related changes down to about 35 km in WACCM-D and 50 km in MLS observations. This point deserves more discussion because the evaluation of the direct particle impact at these altitudes is very important. You correctly highlighted the issue of the MLS observations i.e., OH data become noisier in the stratosphere. Therefore, you cannot accurately evaluate the SPE thresholds in this region by using MLS HOx. Nevertheless, you could potentially do it with other MLS products (e.g., chlorine species) which are properly simulated by WACCM (Funke et al.,
30  2011).

We have highlighted this matter in our conclusions and also discuss the use of other species in the detection of EPPs in the revised manuscript. Kalakoski et al. (2020) showed SPE-related increases in WACCM-D $Cl_x$ down to 1 hPa lasting around a week following an event. They also showed a response in $HNO_3$ above 1 hPa with slightly longer duration (10 days), and enhancements lasting 20–30 days following SPEs below the 1 hPa level.

35  Pag. 7, l30-34: Why? Some issues with the data sampling? Perhaps SPE-related effects could be highlighted even better by using geographic latitudes.

We are somewhat unsure what this refers to. There may be an error in the comment, since the lines given as reference are exactly the same as for the previous comment, and the comment does not seem relevant to these lines.

This comment may be in reference to p. 8, lines 4–7 discussing differences in daytime threshold detection between SH and NH. The differences do arise from sampling of the data and the distribution of the magnetic latitudes and the geographic latitude coverage of the MLS measurements (82°S to 82°N (Waters et al., 2006)). These together lead to the daytime magnetic latitude bin 80°N having more than 35% more MLS measurement points than the 80°S bin. This is likely to cause the discrepancy between NH and SH in daytime SPE detection. Hence this is an issue of data availability at very high latitudes but does not mean that geographic latitudes are necessarily better suited to the study of SPE effects. Using geographic latitudes would not increase the amount of observations available for analysis, the data would simply be binned in a different manner. We believe the geomagnetic latitudes are a better choice, especially where the latitudes are evenly covered by the measurements, because $HO_x$ impact is clearly seen along the geomagnetic latitudes (e.g. Andersson et al., 2014).

**3    Methods**

First, we present an overall comparison between WACCM-D and MLS odd hydrogen data. We use monthly average concentrations to identify the main similarities and differences between the data sets, focusing on the shape, strength, and location of concentration peaks.

15  These comparisons provide us information on the overall representation of odd hydrogen in WACCM-D, which can help to understand the differences in EPP detection thresholds.

Starting from the approach of Verronen et al. (2011), we use daily average data in EPP detection. However, Verronen et al. (2011) studied only four selected RBE events. And although they demonstrated the correlation between EPP and OH, they

20 did not use $HO_2$ data or pursue the detection limits. Before determining threshold values for the SPE and RBE detection, daily climatologies of OH and $HO_2$ concentrations were removed from the odd hydrogen data. Climatological values were calculated for each day of the year by first calculating daily average OH and $HO_2$ night and day concentrations of available data at each grid point, separately for WACCM-D and MLS. A nine-day moving average was then calculated to smooth the time series and produce the final climatologies. After the climatology is removed, the background concentrations do not display

25 seasonal variability, which makes it possible to combine EPP event periods from different seasons for the threshold detection. Examples of this de-seasonalising effect can be seen in Fig. 2.

 It should be noted that after the climatology is removed, the $HO_x$ concentrations still have variability from sources other than EPP, and these are included in the data in our analysis. For example, sudden stratospheric warmings (SSWs) contribute to the year-to-year OH variability in the NH middle atmosphere (Damiani et al., 2010b). As our aim is to determine

30 the general threshold values for SPE and RBE detection, we do not separate SSWs years in any particular manner. However, we consider SH and NH separately and only NH is regularly affected by SSWs.

For the SPE threshold determination, we selected data from all months during which the daily proton flux indicates an event, i.e. it exceeds the limit of 10 $protons\,cm^{-2}s^{-1}sr^{-1}$, see Fig. 3 (upper panel). All months used in the analysis are listed in

Table 1. Using data from the selected months,  the SPE flux values  against the climatology-free odd hydrogen concentrations (i.e. anomalies).  Then a first degree polynomial was  fitted to the  HO$_x$ anomalies and the square root of the SPE fluxes. A limit for a significant SPE-driven enhancement in odd hydrogen concentration was calculated by adding  the standard deviation of the concentrations to the median concentration. Since the concentrations are de-seasonalised, the median is expected to be close to zero. The addition of  the standard deviation  suggests that a concentration anomaly of one standard deviation from the climatology is  significant and should be detected. We then find the SPE indicator value at the intersection of the limit and the linear fit; this is the detected SPE threshold flux value.  Examples of the SPE threshold determination are shown in Fig. 4 (left panel). This process is applied for each latitude bin and at each pressure level separately. To identify those thresholds that are reasonable, they are filtered using correlations between the square root of the SPE fluxes and odd hydrogen concentrations. All thresholds with corresponding correlation coefficient $\geq 0.35$ are accepted as reasonable and the rest are discarded.  This filtering effectively removes threshold values at lower latitudes where SPE impact is not expected.

The RBE threshold values are determined using the same method but with different RBE flux and correlation limits. RBE months were found using an RBE flux limit of $1.5 \times 10^4$ electrons cm$^{-2}$s$^{-1}$sr$^{-1}$ (see Fig. 3, lower panel), and data from these months are used in the analysis. However, the months having an SPE event, as defined in the previous paragraph, were excluded. The RBE fluxes from MEPED are not reliable  during SPEs (Rodger et al., 2010a), and SPE forcing would likely interfere with the RBE threshold detection. Indeed, exclusion of SPE months leads to stronger correlations between the RBE flux and odd hydrogen concentrations (not shown). For a full list of the months used in the analysis, refer to Table 1. As with SPEs, the RBE thresholds were found using  linear fitting with square roots, as well as the median and  the standard deviation of odd hydrogen data (see Fig. 4, right panel). For RBE events, threshold values with corresponding flux-concentration correlation  coefficients $\geq 0.25$ are accepted as reasonable.

As seen in Fig. 4 (left panel) at 0.1–0.2 protons cm$^{-2}$s$^{-1}$sr$^{-1}$, in some cases there is a large number of data points at low fluxes. To find out the impact to our analysis, we performed tests where we excluded these low-flux points. In general, correlations between flux data and HO$_x$ concentrations get stronger, but thresholds increase due to larger standard deviation. These effects would, however, not change our conclusions.

**4   Results**

**4.1   Overall comparison between WACCM-D and MLS**

In general, WACCM-D and MLS compare reasonably well in the magnitude and spatio-temporal variability of OH and HO$_2$. In both MLS and WACCM-D daytime concentration profiles, there is a maximum in the stratosphere and mesosphere, which reflects the production being dependent on atomic oxygen and Lyman-alpha radiation, respectively.  Figure 5 shows a time series of monthly average daytime concentrations at magnetic latitude 70°N, and these peaks and the seasonality of HO$_x$ concentrations are clear in both MLS and WACCM-D

latitudes, as seen in Fig. ?? which shows a typical case (April 2009). There are general differences as well. At the equator, the mesospheric concentration peaks are typically stronger in MLS data, and for also the peak altitude is higher by around 2.5 than in WACCM-D. On the other hand, the stratospheric peaks are stronger in WACCM-D, at least based on concentrations . The noisiness of MLS data is clear at lower atmospheric levels, as seen in Fig. ??. At the polar regions, where EPP impact is

5    expected, absolute differences are smaller while WACCM-D shows generally larger OH concentrations. Thus, in this month, the EPP detection from MLS OH data could be easier due to the lower background concentration in the mesosphere. Overall the greatest concentrations are seen in the summer months. Latitude-wise, the strongest concentrations are found at lower latitudes with concentrations generally decreasing polewards (not shown). Lower latitudes also show the clearest stratospheric and mesospheric maxima and less seasonal variability than the polar regions.

10    The daytime similarities and differences are also visible in the time series of monthly average concentrations , shownfor magnetic latitude 70°N in Fig. 5. Maximum peak values are seen in the summer months in both WACCM-D and MLS. WACCM-D shows a stronger stratospheric OH peak by 10–20% and a weaker mesospheric OH peak by up to a factor of two. In WACCM-D, the vertical transition between the OH summertime peaks is more continuous, while in MLS data there is a clear minimum between them around 0.1 hPa. EPP detection in the summer mesosphere is likely harder from MLS than

15    WACCM-D because of larger background concentrations. Again, the MLS In wintertime, both WACCM-D and MLS show lower OH values than in summer, particularly in the mesospheric altitudes, while the altitude distributions and concentrations are very similar. Similar observations can be made in daytime $HO_2$ data become noisy at the as well (Fig. 5). Of the two maxima in the concentration profiles, the mesospheric one in stronger in MLS $HO_2$ compared to WACCM-D, and MLS also has a clearer minimum between the two maxima at around 0.1 hPa. However the noisiness of MLS $HO_2$ data is evident at

[revised manuscript text omitted]

Overall, we detect RBE only at latitudes poleward of $60°$ (NH and SH) and at altitudes from roughly 65 to 75 km. In WACCM-D, the detection threshold is mostly 1.05–2.55 $\times 10^4$ electrons $\mathrm{cm}^{-2}\mathrm{s}^{-1}\mathrm{sr}^{-1}$, with lower values from WACCM-D when using the $HO_2$ data. With MLS OH data, in NH the detected RBE threshold values are  similar to those with WACCM-D OH, but with a slightly larger spread, i.e. 1.00–2.95 $\times 10^4$ electrons $\mathrm{cm}^{-2}\mathrm{s}^{-1}\mathrm{sr}^{-1}$, and cover

a wider altitude range from 60 to 80 km. MLS OH seems to show a wider latitude range for detection in the correlations as well, extending over latitudes $50°$–$70°$, although RBE can be detected only in one grid point outside $60°$N. A possible indication of this wider latitude range can also be seen in SH in WACCM-D HO$_2$, where a single threshold value can be detected at $70°$S. No thresholds can be found for MLS HO$_2$ nighttime data, i.e. the situation is the same as for the SPE detection. There are no

5  clear correlations between MLS HO$_2$ and RBE indicator (not shown). In daytime, all correlations between the RBE indicator and HO$_x$ concentrations are low. The number of detected daytime RBE indicator threshold values is only  two, both with MLS OH (Fig. 11). These thresholds are 2.16 and 2.50 $\times 10^4$ electrons cm$^{-2}$s$^{-1}$sr$^{-1}$, at altitudes 0.0147 hPa and 0.0215 hPa, respectively, both at magnetic latitude $70°$N.

In an attempt to improve some of the results, five-day averaged data were also examined. A moving five-day average was
10  calculated from the HO$_x$ data and analysed as above with the SPE and RBE indicators. This was done to remove some of the noise, especially in the MLS HO$_2$, but the results were not improved. The data smoothing effectively flattened out the daily concentration peaks caused by events, which generally led to slightly lower correlations with the SPE and RBE indicators. Thus the daily values can be considered an optimal choice for EPP detection, taking into account that a typical SPE/RBE event duration is days.

15  Although RBE forcing in WACCM-D is applied at geomagnetic latitudes $40°$–$72°$, the detection in HO$_x$ impact is only seen at $55°$–$65°$. Unlike the SPE forcing, RBE is not uniform over the latitude range but peaks at the heart of the outer radiation belt (van de Kamp et al., 2016). Thus, only this region  can be used to detect HO$_x$ impact. In the MLS data, i.e. in the  correlations shown in Fig. 10, the RBE extent in the NH seems to reach into neighboring bins outside $55°$–$65°$ although the correlation limit is not exceeded. This could indicate an underestimation in WACCM-D RBE forcing which is
20  driven by the geomagnectic $Ap$ index.

The RBE threshold fluxes are of the order of $10^4$ electrons cm$^{-2}$s$^{-1}$sr$^{-1}$, i.e. 100 times larger fluxes than for SPEs. This is consistent considering that 100 keV electrons ionize about 100 times less molecules than 10 MeV protons while penetrating to about the same atmospheric altitude. Considering the time period 2004–2018, there are 192 days, i.e. 3.9% of all days, which have RBE flux larger than the $10^4$ threshold. Note that our RBE thresholds are an order of magnitude larger than
25  those given by Verronen et al. (2011). Analyzing four large RBE events, they estimated that it is not possible to detect HO$_x$ impact from electron forcing less than 10–30 counts/s as measured by MEPED, and this count rate corresponds to fluxes of $1 - 3 \times 10^3$ electrons cm$^{-2}$s$^{-1}$sr$^{-1}$ (Evans and Greer, 2004).

**5  Conclusions**

In this study we have used atmospheric HO$_x$ data as a detector of EPP impact in the mesosphere and stratosphere. In a sense,
30  WACCM-D simulations have provided us with the theoretical thresholds for the detection, while MLS observations are the present reality that is affected also by the quality of the measurements.

Overall, SPE impacts can be well detected using average nighttime OH data from MLS. Based on the WACCM-D results, detection should be possible also at daytime and using HO$_2$ data. In practise, however, the current MLS data do not have good

enough signal-to-noise ratio to do this. RBE detection is possible as well, but only at nighttime and for more limited altitude and latitude ranges.  While the SPE impact can be seen rather uniformly poleward of $60°$, the RBE impact is focused at $60°$. As with SPEs, only MLS OH observations can be used for confident RBE detection on a day-by-day basis.

Our analysis shows the extent of SPE atmospheric impacts, down to around 35 km and 50 km in WACCM-D and MLS, respectively. The noise in MLS observations likely causes the difference in the altitude extent. Below 50 km altitudes, MLS $HO_x$ can not be reliably used to detect SPEs. Potentially, however, other species like $Cl_x$ or $HNO_3$ could provide better SPE detection capabilities in the stratosphere. For example, the simulations of SPE impacts by Kalakoski et al. (2020) showed enhancements in $Cl_x$ and $HNO_3$ between 1 and 0.01 hPa lasting around a week, and longer lasting (20–30 days) effects below the 1hPa level in $HNO_3$.

We find thresholds for EPP detection using the GOES $>10$ MeV proton fluxes and POES 100–300 keV electron fluxes   to be around $50 - 130$ protons $cm^{-2}s^{-1}sr^{-1}$ and $1.0 - 2.5\times10^4$ electrons $cm^{-2}s^{-1}sr^{-1}$ at nighttime. These flux values have to be exceeded to cause detectable $HO_x$ impact. This limits the data usability to relatively large events. Note, however, that this does not mean that EPP with smaller fluxes is insignificant for the atmosphere. If applied for longer periods of time, EPP below the threshold limit can cause cumulative impacts on chemically long-lived species like $NO_x$.

Although the MLS $HO_2$ data were found to be too noisy for day-to-day EPP detection, they still have a great potential for other purposes. For example, studies of solar-cycle variability in the mesosphere could greatly benefit from the long timeseries. Also, it has been shown, e.g. by Jackman et al. (2014), that the MLS $HO_2$ data are useful when largest solar proton events are studied.

*Code and data availability.* MLS data are available from the NASA Goddard Space Flight Center Earth Sciences (GES) Data and Information Services Center (DISC, https://mls.jpl.nasa.gov/data). All model data used are available from corresponding author by request. CESM source code is distributed through a public subversion code repository (http://www.cesm.ucar.edu/models/cesm1.0/)

*Competing interests.* Authors declare that no competing interests are present.

*Acknowledgements.* The authors would like to thank the CHAMOS group (http://chamos.fmi.fi) for useful discussions.

[revised manuscript text omitted]

**Average monthly daytime concentrations, mlat 70**

[Figure]

**Figure 5.** Monthly average daytime OH and $HO_2$ concentrations on magnetic latitude $70°N$. From top to bottom: WACCM-D OH; MLS OH; WACCM-D $HO_2$; MLS $HO_2$. Concentrations are in units $10^6$ $cm^{-3}$.

[Figure]

**Figure 6.** Monthly average nighttime OH and HO$_2$ concentrations on magnetic latitude $60°$S. From top to bottom: WACCM-D OH; MLS OH; WACCM-D HO$_2$; MLS HO$_2$. Concentrations are in units $10^6$ cm$^{-3}$.

[Figure]

**Figure 7.** Nighttime SPE indicator thresholds (left) and corresponding correlations (right). WACCM-D $HO_2$ (top), WACCM-D OH (middle), and MLS OH (bottom).

[Figure]

**Figure 8.** Daytime SPE indicator thresholds (left) and corresponding correlations (right). WACCM-D HO$_2$ (top), WACCM-D OH (middle), and MLS OH (bottom).

[Figure]

**Figure 9.** Correlations between SPE indicator and MLS HO$_2$ concentrations, (left) daytime and (right) nighttime. Due to the low correlations no threshold values could be detected.

[Figure]

**Figure 10.**  Nighttime RBE indicator thresholds (left) and corresponding correlations (right). WACCM-D  HO2 (top), WACCM-D OH (middle), and  MLS  OH (bottom).

[Figure]

**Figure 11.** Daytime RBE indicator thresholds (left) and corresponding correlations (right) for MLS OH.

**Table 1.** Months included in the analysis, with dates of SPE and RBE flux peaks within each month also given. For peaks within five days of each other, only the date of the strongest is included. Dates marked by * indicate dates in months, where the event peak was in another month. RBE peak dates marked by × are not included in the analysis, due to SPE peak in the same month.

| month | SPE  peaks | RBE  peaks | month | SPE  peaks | RBE  peaks | month | SPE  peaks | RBE  peaks |
|---|---|---|---|---|---|---|---|---|
| 08/2004 | — | 31.08.2004 | 08/2011 | 05.08.2011 | — | 07/2015 | — | 13.07.2015 |
| 09/2004 | 14.09.2004 | — | 09/2011 | 26.09.2011 | — | 08/2015 | — | 17.08.2015 |
| | 20.09.2004 | | 11/2011 | 27.11.2011 | — | | | 27.08.2015 |
| 11/2004 | 01.11.2004 | — | 01/2012 | 24.01.2012 | — | 09/2015 | — | 11.09.2015 |
| | 08.11.2004 | | | 30.01.2012 | | | | 20.09.2015 |
| 01/2005 | 17.01.2005 | 02.01.2005× | 03/2012 | 08.03.2012 | — | 10/2015 | — | 08.10.2015 |
| 03/2005 | — | 07.03.2005 | | 14.03.2012 | | 11/2015 | — | 04.11.2015 |
| 04/2005 | — | 05.04.2005 | 04/2012 | — | 25.04.2012 | 12/2015 | — | 20.12.2015 |
| 05/2005 | 15.05.2005 | 08.05.2005× | 05/2012 | 17.05.2012 | — | | | 31.12.2015 |
| | | 16.05.2005× | 07/2012 | 07.07.2012 | 15.07.2012× | 01/2016 | — | 21.01.2016 |
| | | 30.05.2005× | | 13.07.2012 | | 02/2016 | — | 16.02.2016 |
| 06/2005 | 17.06.2005 | 12.06.2005× | | 19.07.2012 | | 04/2016 | — | 14.04.2016 |
| 07/2005 | 15.07.2005 | 12.07.2005× | 09/2012 | 02.09.2012 | — | 05/2016 | — | 08.05.2016 |
| | 29.07.2005 | | 10/2012 | — | 13.10.2012 | 09/2016 | — | 02.09.2016 |
| 08/2005 | 23.08.2005 | 25.08.2005× | 03/2013 | — | 17.03.2013 | | | 29.09.2016 |
| | | 31.08.2005× | 04/2013 | 11.04.2013 | — | 10/2016 | — | 13.10.2016 |
| 09/2005 | 10.09.2005 | — | 05/2013 | 17.05.2013 | 01.05.2013× | | | 25.10.2016 |
| 12/2005 | — | 11.12.2005 | | 23.05.2013 | | 12/2016 | — | 21.12.2016 |
| 01/2006 | — | 26.01.2006 | 06/2013 | — | 07.06.2013 | 03/2017 | — | 02.03.2017 |
| 03/2006 | — | 19.03.2006 | | | 29.06.2013 | | | 27.03.2017 |
| 04/2006 | — | 14.04/2006 | 09/2013 | 30.09.2013* | — | 04/2017 | — | 22.04.2017 |
| 11/2006 | — | 30.11.2006 | 10/2013 | 01.10.2013 | — | 05/2017 | — | 20.05.2017 |
| 12/2006 | 07.12.2006 | — | 01/2014 | 08.01.2014 | — | | | 28.05.2017 |
| | 13.12.2006 | | 02/2014 | 27.02.2014 | — | 07/2017 | — | 17.07.2017 |
| 05/2007 | — | 23.05.2007 | 03/2014 | 01.03.2014* | — | 08/2017 | — | 23.08.2017 |
| 02/2008 | — | 29.02.2008 | 04/2014 | 19.04.2014 | — | 09/2017 | 05.09.2017 | 28.09.2017 |
| 03/2008 | — | 27.03.2008 | 09/2014 | 12.09.2014 | — | | 11.09.2017 | |
| 04/2010 | — | 06.04.2010 | 12/2014 | — | 07.12.2014 | 10/2017 | — | 13.10.2017 |
| 05/2010 | — | 02.05.2010 | 01/2015 | — | 04.01.2015 | | | 24.10.2017 |
| | | 29.05.2010 | 03/2015 | — | 02.03.2015 | 11/2017 | — | 08.11.2017 |
| 08/2010 | — | 04.08.2010 | | | 18.03.2015 | 04/2018 | — | 20.04.2018 |
| 03/2011 | 08.03.2011 | 11.03.2011× | 04/2015 | — | 16.04.2015 | 05/2018 | — | 06.05.2018 |
| 05/2011 | — | 28.05.2011 | 05/2015 | — | 13.05.2015 | 08/2018 | — | 26.08.2018 |
| 06/2011 | 07.06.2011 | — | 06/2015 | 21.06.2015 | 23.06.2015× | | | |